# Behavior-Discriminative Reward Shaping for Reward-Robust Reinforcement Learning

## Abstract

Reward-robust RL typically models misspecification by specifying an uncertainty set that constrains the discrepancy of plausible rewards from a base reward. However, such reward space discrepancies can be behaviorally irrelevant: under potential-based reward shaping (PBRS), many distinct rewards preserve policy ordering. This can introduce substantial redundancy into standard uncertainty sets and degrade optimization performance. We propose **Shaping-Aware Reward-Robust RL**, which constructs uncertainty sets over PBRS equivalence classes by projecting each reward to a canonical representative, ensuring that the resulting set contains only rewards that induce behaviorally distinct policy rankings. We prove that this projection preserves the optimal robust value while shrinking the uncertainty set and improving empirical performance. Using an extension of the robustness-regularization correspondence, we obtain a practical algorithm to solve the shaping-aware reward-robust RL problem and enjoy convergence guarantees under standard assumptions. Experiments on both continuous-control and discrete grid-world tasks show consistent improvements over representative robust RL baselines and exhibit improved robustness to reward perturbations.

## 1. Introduction

Despite the impressive success of reinforcement learning (RL) across a wide range of domains (Kiran et al., 2021; Christiano et al., 2017; Kober et al., 2013; Liu et al., 2024), deploying RL in real-world systems remains challenging. A central bottleneck is reward misspecification: the reward function used for training is often an imperfect proxy for the

desired objective due to limited data, human biases, modeling errors, or distribution shift (Mannor et al., 2007; Jeon et al., 2020; Enders et al., 2024). Therefore, in safety and reliability critical settings, we would like policies that continue to perform well under plausible reward perturbations. This motivates reward-robust RL, which models misspecification by assuming all plausible rewards lie within a given uncertainty set and solves a max-min objective that maximizes worst-case performance. Equivalently, it can also be viewed as a dynamic zero-sum game: an adversary selects the worst-case reward within the uncertainty set, while the agent seeks a policy that maximizes the resulting worst-case return (Morimoto & Doya, 2005; Lim et al., 2013; Pinto et al., 2017; Liu et al., 2025b).

Existing reward-robust RL frameworks typically construct uncertainty sets by measuring the expected divergence of plausible rewards from a base reward (Gadot et al., 2024; Eysenbach & Levine, 2021). However, a key weakness that has been largely overlooked in the robust RL literature is that controlling such divergences in reward space does not necessarily correspond to differences in decision-making. In particular, potential-based reward shaping (PBRS), which encompasses a general and widely adopted family of shaping transformations (Ng et al., 1999; Müller & Kudenko, 2025; Lidayan et al., 2024), shows that certain shaping rewards leave the optimal policy invariant and, more generally, preserve policy ordering. As a consequence, a divergence-based uncertainty set can contain many behaviorally equivalent rewards: they differ under the chosen divergence measure but induce identical policy rankings. This redundancy can be harmful in robust optimization. In particular, it can waste adversary iterations on rewards that provide no new behavioral distinctions rather than finding genuinely more adverse rewards (as theoretically justified and empirically shown in Section 3.1 and Figure 2), ultimately weakening the adversary and degrading robust training.

In this work, we make the first step towards behavior-discriminative robust RL by proposing **Shaping-Aware Reward-Robust RL**. Our framework addresses the above issues by constructing a shaping-aware projected uncertainty set in which each PBRS-equivalent reward class is represented by a canonical reward. Intuitively, this removes

---

[1]Anonymous Institution, Anonymous City, Anonymous Region, Anonymous Country. Correspondence to: Anonymous Author <anon.email@domain.com>.

Preliminary work. Under review by the International Conference on Machine Learning (ICML). Do not distribute.

rewards that provably do not affect policy ranking and yields an uncertainty set that better reflects behaviorally distinct objectives. We show that this reduction does not change the optimal robust value of the original reward-robust RL problem, while substantially simplifying the adversary's search space by reducing dimensions. Empirically, we find that optimization on the projected uncertainty set makes adversary updates more identifiable and improves both performance and convergence.

However, directly solving the max-min robust RL problem with projected rewards requires projecting every candidate reward considered by the adversary to its canonical representative, which is computationally expensive. To address this, we leverage the well-established connection between regularization and robustness in the RL setting (Husain et al., 2021; Eysenbach & Levine, 2021; Derman et al., 2023; Ashlag et al., 2025), which states that optimizing a suitably regularized objective is equivalent to solving a reward-robust RL problem for an associated uncertainty set. We show that this correspondence can be extended to our projected uncertainty set, which allows us to reduce the required projections to only the base reward and simplifies solving the max-min problem. We further choose the canonical representative by minimizing the discrepancy induced by the uncertainty set within each PBRS equivalence class. This projection selects a representative that is consistent across equivalence classes and conservative with respect to the robust objective. Putting these together, we derive a practical algorithm that alternates between learning the canonical form of the base reward and optimizing the corresponding regularized RL objective. Since our approach only introduces a lightweight projection-learning step prior to standard policy updates, it can be integrated with a wide range of existing RL optimizers. Moreover, under standard assumptions, we establish convergence guarantees for the overall algorithm.

We empirically evaluate our method on both discrete and continuous control tasks, comparing against representative robust RL baselines. Across all environments, our method consistently achieves higher returns and reaches strong performance in fewer training steps. We further assess robustness under reward perturbations and observe that our method exhibits smaller performance degradation than standard baselines when rewards are shifted. Overall, these results support the view that removing PBRS-induced redundancy leads to more effective robust policy learning under reward uncertainty.

To summarize, the key contribution of this work is: 1) To the best of our knowledge, we are the first to incorporate PBRS equivalence into the reward-robust RL and to construct uncertainty sets based on behavioral distinctions, rather than raw reward discrepancies, yielding stronger robust learning performance. 2) We derive a practical shaping-aware

reward-robust RL algorithm that trains policies against this behaviorally distinct uncertainty set, and we provide convergence guarantees under standard assumptions. 3) We empirically validate the method across diverse environments, showing higher returns, faster convergence, and improved robustness to reward perturbations compared to representative baselines.

## 2. Preliminary

**Notation.** For a finite set $\mathcal{Z}$, we write $\mathbb{R}^{\mathcal{Z}}$ for the set of real-valued functions on $\mathcal{Z}$, and $\Delta_{\mathcal{Z}}$ for the probability simplex over $\mathcal{Z}$. More generally, given any set $\mathcal{X}$, $\Delta_{\mathcal{Z}}^{\mathcal{X}}$ denotes the collection of mappings from $\mathcal{X}$ into $\Delta_{\mathcal{Z}}$. For $a, b \in \mathbb{R}^{\mathcal{Z}}$, we define the standard inner product as $\langle a, b \rangle := \sum_{z \in \mathcal{Z}} a(z) b(z)$. Let $C \subset \mathbb{R}^{\mathcal{Z}}$ be a convex set and let $\Omega : C \to \mathbb{R}$ be strongly convex. Throughout the paper, $\Omega$ serves as a policy regularizer. We define $\Omega^*(y) = \arg\max_{a \in C} \{\langle a, y \rangle - \Omega(a)\}, \forall y \in \mathbb{R}^{\mathcal{Z}}$, which corresponds to the Legendre-Fenchel transform (also known as the convex conjugate) on $C$. Moreover, $\Omega^*$ is Lipschitz and satisfies: for $\lambda > 0$, the convex conjugate of $\lambda \Omega(\cdot)$ is $\lambda \Omega^*(\frac{\cdot}{\lambda})$ (Hiriart-Urruty & Lemaréchal, 2004; Mensch & Blondel, 2018; Bertsekas, 2009).

**Standard RL.** Consider an infinite-horizon Markov decision process (MDP) $(\mathcal{S}, \mathcal{A}, \mu_0, \gamma, P, r)$, where $\mathcal{S}$ and $\mathcal{A}$ are finite state and action spaces, $0 < \mu_0 \in \Delta_{\mathcal{S}}$ is the initial state distribution and $\gamma \in (0, 1)$ is the discount factor. Define the state action space $\mathcal{X} := \mathcal{S} \times \mathcal{A}$ and the transition space $\mathcal{X}' := \mathcal{S} \times \mathcal{A} \times \mathcal{S}$. The dynamics are given by a transition kernel $P \in \Delta_{\mathcal{S}}^{\mathcal{X}}$, which assigns to each $(s, a) \in \mathcal{X}$ a distribution over next states in $\mathcal{S}$. The reward is a function $r \in \mathbb{R}^{\mathcal{X}'}$ defined over $(s, a, s') \in \mathcal{X}'$, where $s'$ generated from $P(\cdot|s, a)$ is the state reached after executing action $a$ in state $s$. A policy $\pi \in \Delta_{\mathcal{A}}^{\mathcal{S}}$ specifies an action distribution for each state $s \in \mathcal{S}$. Given $\mu_0, \pi$ and $P$, the discounted return under reward $r$ is: $J(\pi, r) := \mathbb{E}\left[\sum_{t=0}^{\infty} \gamma^t r(s_t, a_t, s_{t+1}) \Big| \mu_0, \pi, P\right]$. The standard RL objective is to find a policy $\pi$ that maximizes the discounted return.

**Potential-Based Reward Shaping.** Potential-based reward shaping (PBRS) (Ng et al., 1999) augments the environment reward via a potential function $\Phi \in \mathbb{R}^{\mathcal{S}}$, which assigns a scalar to each state. Given $\Phi$, PBRS defines a shaping function $F$ on transition tuples $(s, a, s') \in \mathcal{X}'$ by $F(s, a, s') = \gamma\Phi(s') - \Phi(s)$, and the shaping reward $r' \in \mathbb{R}^{\mathcal{X}'}$ as $r'(s, a, s') = r(s, a, s') + F(s, a, s')$. A key property of PBRS is policy invariance: replacing $r$ with $r'$ leaves the optimal policy unchanged and, more generally, preserves the return ordering among all policies (Theorem 1 of (Ng et al., 1999)).

**Reward-Robust RL.** A reward-robust MDP is given by

$(\mathcal{S}, \mathcal{A}, \mu_0, \gamma, P, \mathcal{U})$, where the true reward is unknown but assumed to lie in a prescribed uncertainty set $\mathcal{U} \subseteq \mathbb{R}^{\mathcal{X}'}$. Typically, $\mathcal{U}$ is defined around a base reward $r \in \mathbb{R}^{\mathcal{X}'}$ through a discrepancy budget: $\mathcal{U} = \{\tilde{r} \in \mathbb{R}^{\mathcal{X}'} : \Omega^*(r - \tilde{r}) \leq \epsilon\}$, where $\Omega^*$ is a dissimilarity measure and $\epsilon > 0$ controls the robustness level. The reward-robust RL objective seeks a policy $\pi$ that maximizes worst-case return:

$$\max_{\pi \in \Delta_{\mathcal{A}}^{\mathcal{S}}} \min_{\tilde{r} \in \mathcal{U}} J(\pi, \tilde{r}). \tag{1}$$

The uncertainty sets studied in the literature typically arise from specific choices of $\Omega^*$. For example, define the reward difference $u(s, a, s') = r(s, a, s') - \tilde{r}(s, a, s')$, the simplest and common used is $l_p$-constrained uncertainty set, where $\Omega^*(u) = \|u\|_p$ (Kumar et al., 2023; Ho et al., 2021; Kumar et al., 2025). In addition, (Eysenbach & Levine, 2021) consider an $\Omega^*$ of the form $\Omega^*(u) = \mathbb{E}\left[\sum_t \gamma^t \log \sum_a \exp(u(s_t, a, s_{t+1})) \Big| \mu_0, \pi, P\right]$, which induces a log-sum-exp discrepancy over reward differences along trajectories. In addition, (Derman et al., 2023) define a reference policy weighted variant: $\Omega^*(u) = \mathbb{E}\left[\sum_t \gamma^t \log \sum_{a \in \mathcal{A}} \pi_{\text{ref}}(a|s_t) \exp(\frac{u(s_t, a, s_{t+1})}{\tau}) \Big| \mu_0, \pi, P\right]$, where $\pi_{\text{ref}}$ is a reference policy and $\tau > 0$ is a temperature parameter controlling the strength of the discrepancy. Notice that most $\Omega^*$ considered in prior work can be derived from a regularizer $\Omega$ (detailed below). Our work mainly focuses on the uncertainty set induced by such $\Omega^*$.

**Regularized RL.** A regularized MDP is a tuple $(\mathcal{S}, \mathcal{A}, \mu_0, \gamma, P, r, \Omega)$, where $\Omega$ denotes an additional regularization term. The regularized RL objective takes the form

$$\max_{\pi \in \Delta_{\mathcal{A}}^{\mathcal{S}}} J(\pi, r) + \Omega(\pi), \tag{2}$$

A widely used choice is policy-entropy regularization (Haarnoja et al., 2018; Eysenbach & Levine, 2021): $\Omega(\pi) = -\mathbb{E}\left[\sum_t \gamma^t \sum_a \pi(a \mid s_t) \log \pi(a \mid s_t) \Big| \mu_0, \pi, P\right]$. Another common regularizer is the KL divergence to a reference policy (Derman et al., 2023; Filippi et al., 2010): $\Omega(\pi) = -\mathbb{E}\left[\sum_t \gamma^t \sum_a \pi(a \mid s_t) \log \frac{\pi(a|s_t)}{\pi_{\text{ref}}(a|s_t)} \Big| \mu_0, \pi, P\right]$. Prior work has established a connection between regularization and robustness in RL for specific choices of $\Omega$ and $\Omega^*$ (Eysenbach & Levine, 2021; Derman et al., 2023; Husain et al., 2021). In particular, entropy-regularized and KL-regularized RL objectives can be interpreted as reward-robust RL problems, with corresponding discrepancies given by the log-sum-exp penalty (Eysenbach & Levine, 2021) and a reference policy weighted variant (Derman et al., 2023). For simplicity, in the remainder of the paper, we refer to these reward-robust formulations by the name of their corresponding regularizer (e.g., "entropy" or "KL") whenever the meaning is clear. A detailed related work on PBRS, reward-robust RL, and regularized RL is in Appendix A.

## 3. Shaping-aware Reward-Robust RL

Solving the inner minimization in the reward-robust RL objective (1) requires an adversary to search for the worst-case reward within the uncertainty set. However, when the uncertainty set is defined directly in reward space, it can contain many PBRS-equivalent rewards that induce identical policy behavior, leading to redundant adversary search. In Section 3.1, we illustrate this issue on a toy MDP and discuss its impact on optimization behavior. In Section 3.2, we introduce **Shaping-Aware Reward-Robust RL**, which removes this redundancy by projecting the uncertainty set onto PBRS equivalence classes so that each element corresponds to a behaviorally distinct objective. We show that this projection preserves the optimal robust value while yielding a smaller and more identifiable uncertainty set.

### 3.1. Why PBRS redundancy harm optimization?

In this section, we illustrate PBRS-induced redundancy and its optimization implications via the toy MDP in Figure 1a. This MDP has two states $s_1, s_2$ and two actions $a_1, a_2$. From $s_1$, either action deterministically transitions to $s_2$. Let $r_1 := r(s_1, a_1, s_2)$, $r_2 := r(s_1, a_2, s_2)$, and consider the rectangular uncertainty set in Figure 1b (left), where $\mathcal{U} := \{(r_1, r_2) : r_1 \in [0, 1], r_2 \in [0, 1]\}$. We first characterize the PBRS-equivalent rewards within $\mathcal{U}$. As discussed in Section 2, a potential function takes the form $F(s, a, s') = \gamma \Phi(s') - \Phi(s)$. In this toy MDP, both actions share the same transition $s_1 \to s_2$, hence the shaping term applied to each action is identical, i.e., $F_1 = F_2 = \gamma \Phi(s_2) - \Phi(s_1) =: F$. Consequently, adding the same shift to both rewards, $(\tilde{r}_1, \tilde{r}_2) = (r_1, r_2) + F \cdot (1, 1)$, does not change policy ranking. The only behaviorally relevant quantity is the reward difference $\Delta := r_1 - r_2$. Geometrically, PBRS equivalence classes correspond to the slope-1 diagonals in the $(r_1, r_2)$-plane. As illustrated by the dashed lines in Figure 1b (left), all rewards lying on the same dashed diagonal are behaviorally equivalent. As a result, $\mathcal{U}$ contains entire families of distinct rewards that are PBRS equivalent, introducing substantial redundancy. To remove the redundancy, a natural way is to select a canonical representative $r^{\text{core}}$ from each equivalence class, thereby projecting the original reward space onto a lower-dimensional, behaviorally distinct space. Many canonicalization rules are available. For instance, we may fix $r_2^{\text{core}} = 0$ and set $r_1^{\text{core}} = \Delta$, yielding a one-dimensional set $\{(\Delta, 0) : \Delta \in [-1, 1]\}$ (Figure 1b (middle)). Alternatively, we may center rewards by enforcing zero mean, e.g., $r_1^{\text{core}} = \frac{\Delta}{2}$ and $r_2^{\text{core}} = -\frac{\Delta}{2}$, which corresponds to the diagonal line through the origin (Figure 1b (right)).

The PBRS-induced redundancy can lead to inefficient and sometimes misleading worst-case optimization. Consider the adversary's update over $\tilde{r} \in \mathcal{U}$. If the adversary opti-

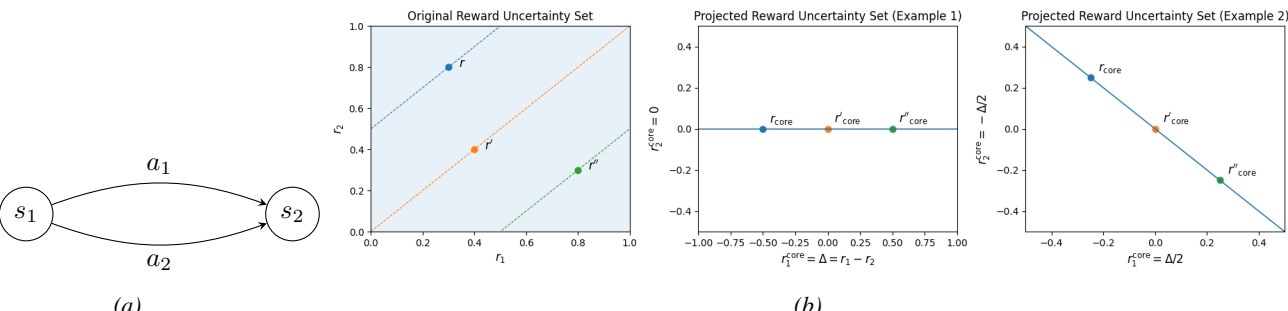

*Figure 1.* (a) A two-state toy MDP. The agent starts in $s_1$ and can choose between two actions, $a_1$ and $a_2$. Regardless of the action taken in $s_1$, the transition to $s_2$ is deterministic. (b) The original reward uncertainty set is the square $[0,1]^2$ over the $(r_1, r_2)$ plane. All points on the same dashed diagonal induce identical policy rankings under PBRS. Projecting each PBRS equivalence class to a canonical representative collapses the uncertainty set to a one-dimensional set. The left figure highlights example rewards $r, r', r''$. The middle and right figure show their corresponding representatives $r_{\text{core}}, r'_{\text{core}}, r''_{\text{core}}$ under two different projection choices.

mizes directly in the original reward space (e.g., via stochastic gradient descent), its iterates can move along PBRS-invariant directions, which do not affect the optimal policy. Though in theory, the adversary would not systematically move in these directions, since the expected gradient along PBRS-invariant directions is zero. In practice, the optimization is stochastic. With finite rollouts and approximate return estimates, gradient estimates can induce non-zero noise even along behaviorally irrelevant directions. This noise can cause a random walk along PBRS-invariant directions, wasting adversary updates and potentially leading to premature stabilization at PBRS-equivalent rewards that provide little genuine progress in lowering the worst-case value. By contrast, optimizing over the projected reward space, where PBRS-invariant directions are removed, ensures that each adversary update is identifiable in the sense that it can meaningfully affect the worst-case objective. As a result, even under noisy gradient estimates, the adversary avoids spending iterations on redundant PBRS-equivalent rewards. We further show this effect in Section 5 and Appendix D.1 in non-convex uncertainty sets.

### 3.2. Shaping-aware Projected Uncertainty Set

In this section, we make the redundancy phenomenon precise and introduce our shaping-aware reward-robust RL framework to address it. We first define behaviorally equivalent rewards under PBRS: rewards that differ only by a PBRS term shift the return of every policy by the same constant, and therefore induce identical behavior in the sense of decision-making (i.e., leave policy ranking unchanged) (Ng et al., 1999).

**Lemma 3.1.** *Let $\Phi \in \mathbb{R}^{\mathcal{S}}$ be any bounded potential function and let $F(s, a, s') = \gamma \Phi(s') - \Phi(s)$ be the corresponding shaping term. Define the shaped reward $r'(s, a, s') = r(s, a, s') + F(s, a, s')$. Then, for any policy $\pi$, $J(\pi, r') = J(\pi, r) + C(\Phi, \mu_0)$, where $C(\Phi, \mu_0) = -\mathbb{E}_{s_0 \sim \mu_0}[\Phi(s_0)]$ depends only on $\Phi$ and the initial distribution $\mu_0$, and is*

*independent of $\pi$. We say that $r'$ is PBRS-equivalent to $r$.*

Lemma 3.1 is a well-known property of PBRS and we include the proof in Appendix B.1 for completeness. Since the additive term $C(\Phi, \mu_0)$ does not affect comparisons between policies, without loss of generality, we may restrict attention to the potential function $\Phi$ satisfying $C(\Phi, \mu_0) = 0$, and we adopt this convention throughout the remainder of the paper. Next, we show that the standard uncertainty sets considered in the literature include rewards that are behaviorally indistinguishable under PBRS.

**Proposition 3.2.** *Let $\mathcal{U}$ be an uncertainty set as defined in Section 2, of the form $\mathcal{U} = \{\tilde{r} : \Omega^*(r - \tilde{r}) \le \epsilon\}$. Assume $\mathcal{U}$ contains a neighborhood of some $\tilde{r}$ (e.g., $\mathcal{U}$ has nonempty interior around $\tilde{r}$). Then there exists a nontrivial shaping direction $F \neq 0$ (generated by some $\Phi \in \mathbb{R}^{\mathcal{S}}$) and a scalar $\delta > 0$ such that $\tilde{r} + \alpha F \in \mathcal{U}$, for all $|\alpha| \le \delta$. All rewards in $\{\tilde{r} + \alpha F : |\alpha| \le \delta\}$ are PBRS-equivalent and therefore induce identical policy rankings.*

The proof is provided in Appendix B.2. Proposition 3.2 highlights that, because $\mathcal{U}$ is defined over raw rewards, it can contain entire neighborhoods of PBRS-equivalent rewards. This redundancy motivates constructing an uncertainty set over PBRS equivalence classes, so that each element represents a unique, behaviorally distinct objective. To formalize this idea, let $\mathcal{R} \subseteq \mathbb{R}^{\mathcal{X}'}$ denote the reward space of interest. Define the shaping subspace $\mathcal{S}_{\text{shape}} := \{F_\Phi : \Phi \in \mathbb{R}^{\mathcal{S}}\}$, and the corresponding quotient space of PBRS equivalence classes $\mathcal{R}_{\text{core}} := \mathcal{R}/\mathcal{S}_{\text{shape}}$, whose elements are classes $[r] = \{r + F_\Phi : F_\Phi \in \mathcal{S}_{\text{shape}}\}$. Assume we fix a canonical representative for each equivalence class via a projection map $\text{Proj} : \mathcal{R} \to \mathcal{R}$ such that for any $r \in \mathcal{R}$, $\text{Proj}(r) \in [r]$ and $\text{Proj}(r + F_\Phi) = \text{Proj}(r)$ for all $F_\Phi \in \mathcal{S}_{\text{shape}}$. Using the above definition, given any uncertainty set $\mathcal{U} \subseteq \mathcal{R}$, we then define its projected uncertainty set as $\mathcal{U}_{\text{core}} := \text{Proj}(\mathcal{U}) = \{\text{Proj}(r) : r \in \mathcal{U}\} \subseteq \mathcal{R}$, which collapses PBRS-equivalent rewards in $\mathcal{U}$ to a single representative defined by Proj. We now establish that

working with $\mathcal{U}_{\text{core}}$ is not only conceptually cleaner, but also preserves the robust objective. In particular, we have:

**Theorem 3.3.** *Let $\mathcal{U}_{\text{core}}$ be the projected uncertainty set of $\mathcal{U}$ as defined above, we have*

$$\max_{\pi \in \Delta_{\mathcal{A}}^{\mathcal{S}}} \min_{\tilde{r} \in \mathcal{U}} J(\pi, \tilde{r}) = \max_{\pi \in \Delta_{\mathcal{A}}^{\mathcal{S}}} \min_{\tilde{r}_{core} \in \mathcal{U}_{core}} J(\pi, \tilde{r}_{core}). \quad (3)$$

*Consequently, the standard reward-robust RL problem can be solved over the shaping-aware projected uncertainty set $\mathcal{U}_{core}$ without changing the robust optimal value.*

The proof of Theorem 3.3 is provided in Appendix B.3. It follows directly from the definition of $\mathcal{U}_{\text{core}}$ together with Lemma 3.1. Next, we quantify how projection reduces the adversary's effective search space. Since $\mathcal{U}_{\text{core}}$ collapses all PBRS-equivalent rewards to a single representative, it removes entire directions in reward space that provably do not affect policy ranking. Under a linear parameterization, this reduction can be characterized explicitly as a decrease in affine dimension:

**Proposition 3.4.** *Assume a linear parameterization of the reward function: $r_\theta(s, a, s') = \phi(s, a, s')^\top \theta, \theta \in \Theta$, where $\Theta \subseteq \mathbb{R}^d$ is a convex ambiguity set in parameter space corresponding to $\mathcal{U}$ (i.e, $\mathcal{U} = \{r_\theta : \theta \in \Theta\}$), and let $\mathcal{S}_{shape} \subseteq \mathbb{R}^d$ be the shaping subspace. Let $\text{aff}(\Theta)$ denote the affine hull of $\Theta$. Then the projected set can be carried out over the lower-dimensional set $\Theta_{core} = \{Proj(\theta) : \theta \in \Theta\}$, whose affine dimension is at most $\dim(\Theta_{core}) \leq \dim(\Theta) - \dim(\mathcal{S}_{shape} \cap \text{aff}(\Theta))$. In particular, whenever $\mathcal{S}_{shape}$ is nontrivial and intersects the span of $\Theta$, the projected search space strictly shrinks.*

The proof is provided in Appendix B.4. In general, replacing $\mathcal{U}$ with $\mathcal{U}_{\text{core}}$ reduces the adversary's search space, and the extent of this reduction scales with the number of independent shaping directions present in $\mathcal{U}$. The toy example in Figure 1 is a concrete illustration: here $\dim(\mathcal{U}) = 2$ and the shaping subspace $\mathcal{S}_{\text{shape}} = \text{span}\{(1, 1)\}$ has dimension 1, so $\mathcal{U}_{\text{core}}$ is one-dimensional. We further show that for any discounted multi-armed bandit, the projection removes exactly one degree of freedom (Appendix B.5). At the other extreme, if $\mathcal{U}$ consists only of shaping rewards, then $\mathcal{U}_{\text{core}}$ collapses to a single representative and removes all dimensions. Additionally, Appendix B.6 shows that under mild assumptions, $\mathcal{U}_{\text{core}}$ inherits key geometric properties of $\mathcal{U}$, such as compactness and convexity. These properties ensure the optimization on $\mathcal{U}_{\text{core}}$ is well-posed.

## 4. Shaping-aware Reward-Robust Algorithm

A direct optimization of (3) requires solving the inner minimization over $\mathcal{U}_{\text{core}}$. In practice, this means maintaining feasibility after each adversary update and thus projecting intermediate rewards back onto $\mathcal{U}_{\text{core}}$ at every inner step.

However, computing this projection is itself a nontrivial optimization, adding substantial per-step overhead. This overhead can be prohibitive in large-scale RL, where the inner minimization is already the main computational bottleneck. While alternative implementations may reduce this cost, designing an efficient algorithm is nontrivial. In Section 4.1, we address the issue by leveraging the robustness-regularization duality and then specify a concrete canonicalization rule. We present the overall algorithm in Section 4.2.

### 4.1. Shaping-aware Projection

We first show that the robustness-regularization duality holds for our projected uncertainty set.

**Theorem 4.1.** *Fix $\epsilon \geq 0$ and $\lambda > 0$. Let $\Omega^*$ denote the discrepancy measure and $\Omega$ be the corresponding regularization term discussed in Section 2. Define the uncertainty set $\mathcal{U} := \left\{\tilde{r} : \Omega^*\left(\frac{r - \tilde{r}}{\lambda}\right) \leq \epsilon\right\}$ and the projected uncertainty set $\mathcal{U}_{core} = Proj(\mathcal{U})$. Let $r_{core} := Proj(r)$ be the projection of base reward $r$. Under Assumption B.5, the shaping-aware reward-robust objective satisfies*

$$\max_\pi \min_{\tilde{r}_{core} \in \mathcal{U}_{core}} J(\pi, \tilde{r}_{core}) = \max_\pi J(\pi, r_{core}) - \lambda\Omega(\pi).$$

The proof is deferred to Appendix B.7, where we extend Theorem 4.1 of (Eysenbach & Levine, 2021) to a general robustness-regularization equivalence, and then show it continues to hold for the projected uncertainty set. Theorem 4.1 implies that we can solve the objective (3) by optimizing a standard regularized objective parameterized by the projected base reward $r_{\text{core}}$. It suffices to only compute $r_{\text{core}}$, substantially reducing computational overhead.

It remains to specify the projection operator $\text{Proj}$, i.e., how we choose a canonical representative from each PBRS equivalence class. In principle, many canonicalizations are possible. Recall that our robust objective measures reward uncertainty using the dissimilarity induced by $\Omega^*$. It is therefore natural to define the projection in a way that is consistent with this geometry. We adopt a conservative canonicalization: within each equivalence class $[r]$, we select the representative that is smallest under the same $\Omega^*$-induced geometry: $\text{Proj}(r) = \arg\min_{r' \in [r]} \Omega^*(r')$. Intuitively, this picks the least inflated reward among all PBRS-equivalent variants, yielding a consistent and geometry-aligned representative across classes. Moreover, this choice avoids inadvertently interpreting PBRS-equivalent rewards outside the original budgeted set as admissible, which would correspond to certifying robustness with respect to a larger uncertainty set than the one we specify. Additional properties of this projection are detailed in Appendix B.8. In particular, we show that the projection is well-defined by establishing existence and uniqueness. We also show that, from a max-min viewpoint, this projection can tighten con-

vergence bounds for certain optimization methods.

To implement $\text{Proj}(r)$, notice that $r' \in [r]$ means that $r$ differs from $r'$ by a potential-based shaping term, i.e., $r = r' + F_\Phi$ with $F_\Phi(s, a, s') = \gamma\Phi(s') - \Phi(s)$. In practice, we approximate $\Phi$ using a neural network $\Phi_\varphi$ with parameters $\varphi$ as a flexible and general function approximation. Then the projection reduces to optimizing over $\varphi$: $\varphi^* = \arg\min_\varphi \Omega^*\left(r - F_{\Phi_\varphi}\right)$. Given the learned parameters $\varphi^*$, we compute the shaping term for each reward as $F_{\Phi_{\varphi^*}}$, and obtain the corresponding projected reward by subtracting this shaping component, $r' = r - F_{\Phi_{\varphi^*}}$.

### 4.2. Shaping-aware Reward-Robust Policy Gradient

We summarize the overall training procedure to solve our shaping-aware reward-robust RL problem in Algorithm 1. At each iteration, we first learn the projected reward by estimating the projection loss $l(\varphi_{i,k}, \theta_i; \mathcal{D}_{\pi_{\theta_i}})$, where $\Omega^*_{\mathcal{D}_{\pi_{\theta_i}}}$ denotes the estimate of $\Omega^*$ computed from the rollout data $\mathcal{D}_{\pi_{\theta_i}}$, and updating $\varphi$ via gradient descent (lines 6-8). We then replace the rewards in the same batch with the resulting projected reward and perform a standard policy update on the regularized objective (lines 10-11). We justify in Appendix B.9 that such on-policy projection-learning design is data-efficient and supports online adaptation. Although Algorithm 1 updates the policy via policy gradients, in principle, our projection simply replaces the environment reward with its canonicalized version, so it is compatible with any RL algorithm that optimizes expected return (e.g., SAC (Haarnoja et al., 2018) or PPO (Schulman et al., 2017)). In Appendix B.9, we prove convergence under standard conditions by first analyzing the projection-learning updates and then applying a two-time scale argument (Bertsekas, 2009; Gadot et al., 2024) to establish convergence of the overall algorithm.

---

**Algorithm 1** Shaping-aware Policy Gradient

---

1: **Hyperparameters:** Iteration number $I$, $K$, Step size $\{\beta_i\}_{i=0}^{I-1}$ for $\theta$, Step size $\{\alpha_{i,k}\}_{i=0,\dots,I-1;\, k=0,\dots,K-1}$ for $\varphi$.
2: **Initialize:** $\theta_0 \in \Theta, \varphi_0 \in \Psi$
3: **for** iteration $i = 0, \dots, I-1$ **do**
4:     Collect a batch dataset $\mathcal{D}_{\pi_{\theta_i}}$ from the current policy $\pi_{\theta_i}$.
5:     Initialize $\varphi_{i,0} = \varphi_i$
6:     **for** iteration $k = 0, \dots, K$ **do**
7:         Update $\varphi$: $\varphi_{i,k+1} = \varphi_{i,k} - \alpha_{i,k}\nabla_\varphi l(\varphi_{i,k}, \theta_i; \mathcal{D}_{\pi_{\theta_i}})$
        where $l(\varphi_{i,k}, \theta_i; \mathcal{D}_{\pi_{\theta_i}}) = \Omega^*_{\mathcal{D}_{\pi_{\theta_i}}}\left(r - F_{\Phi_{\varphi_{i,k}}}\right)$
8:     **end for**
9:     Set $\varphi_{i+1} := \varphi_{i,K}$
10:    Transform $\mathcal{D}_{\pi_{\theta_i}}$ to $\mathcal{D}'_{\pi_{\theta_i}}$ by replacing the rewards with $r' = r - F_{\Phi_{\varphi_{i+1}}}$.
11:    Update $\theta$ via $\theta_{i+1} = \theta_i + \beta_i\nabla_\theta \bar{J}(\theta_i, \varphi_{i+1}; \mathcal{D}'_{\pi_{\theta_i}})$ where $\bar{J}(\theta_i, \varphi_{i+1}; \mathcal{D}'_{\pi_{\theta_i}}) = J_{\mathcal{D}'_{\pi_{\theta_i}}}(\pi_{\theta_i}, r_{\varphi_{i+1}}) - \Omega_{\mathcal{D}'_{\pi_{\theta_i}}}(\pi_{\theta_i})$
12: **end for**

---

## 5. Experiment

We evaluate the performance and robustness of our shaping-aware reward-robust RL framework across a diverse set of both discrete and continuous control tasks and reward perturbations. Our goal is to test whether our method improves (i) nominal performance during robust training and (ii) robustness to reward misspecification, relative to standard reward-robust RL baselines. We conduct experiments on a MiniGrid environment (Chevalier-Boisvert et al., 2018) for the discrete task, where we adopt a $l_2$ constrained uncertainty set and compare policies trained by directly optimizing the standard max-min objective versus the shaping-aware max-min objective. For the continuous task, we conduct experiments on five environments from the OpenAI Gym MuJoCo suite (Todorov et al., 2012; Brockman et al., 2016): Hopper-v4, Walker2d-v4, HalfCheetah-v4, Humanoid-v4, and Reacher-v4. We study both entropy-based and KL-based reward-robust objectives discussed in Section 2. We compare a standard PPO baseline against our approach, which uses the same PPO optimizer and hyperparameters but augments training with the proposed projection step. Additional implementation details are provided in Appendix C. Additional results, including computational overhead, an ablation study on the sensitivity to data coverage for the projection learning, are presented in Appendix D.

**MiniGrid max-min robust training.** The evaluation result in the MiniGrid environment can be seen in Figure 2. Specifically, Figure 2b reports the worst-case return $J(\pi, r_{\text{worst}})$ across outer iterations for the standard max-min baseline and our shaping-aware method. Both methods improve rapidly at the beginning, but the shaping-aware variant continues to make steady progress and converges to a substantially higher worst-case return. This indicates that optimizing over the PBRS-canonicalized reward space leads to a stronger robust solution under the same uncertainty budget and comparable optimization settings. Figure 2c quantifies, at each outer iteration, what fraction of the adversary's update lies in PBRS-invariant directions. For the standard baseline, a persistent and significant fraction of adversary updates moves along shaping directions that do not alter policy orderings. This arises because the inner minimization is solved only approximately (via limited gradient descent), and the stochastic estimation noise (soft-max policies) together with inner-loop hyperparameter choices (stepsize) can induce drift along PBRS-invariant directions. Such updates are effectively wasted from the perspective of identifying more adverse behaviors and lead to premature stabilization at some PBRS-equivalent rewards that provide little genuine progress in lowering the worst-case value. In contrast, our method operates directly in the projected reward space, where PBRS-invariant directions are removed. As a result, the PBRS-invariant component of the adversary update is near zero, and each update is identifiable in the

*Table 1.* Average cumulative returns and percentage improvement for different methods on various MuJoCo environments. Each policy is evaluated for 100 episodes. The return is in terms of the environment's original reward. We report the mean $\pm$ 95% confidence interval over 5 random seeds. $\Delta_{\text{entropy}}$ and $\Delta_{\text{KL}}$ denote the improvement of our method over the corresponding base method.

| Environment | Robust RL (entropy) | Shaping-aware (entropy) | $\Delta_{\text{entropy}}$ | | Robust RL (KL) | Shaping-aware (KL) | $\Delta_{\text{KL}}$ | |
|---|---|---|---|---|---|---|---|---|
| Hopper-v4 | $2905.12 \pm 133.86$ | $3214.68 \pm 115.60$ | $+309.56$ | ↑ 10.7% | $2882.08 \pm 186.12$ | $3249.23 \pm 131.87$ | $+367.15$ | ↑ 12.7% |
| Walker2d-v4 | $2861.81 \pm 316.32$ | $3385.88 \pm 281.79$ | $+524.07$ | ↑ 18.3% | $2676.94 \pm 263.54$ | $3577.35 \pm 352.55$ | $+900.41$ | ↑ 33.6% |
| HalfCheetah-v4 | $1386.73 \pm 21.84$ | $1625.94 \pm 20.58$ | $+239.21$ | ↑ 17.2% | $1583.30 \pm 35.38$ | $3288.30 \pm 60.48$ | $+1705.00$ | ↑ 107.7% |
| Humanoid-v4 | $463.15 \pm 25.82$ | $505.54 \pm 15.78$ | $+42.39$ | ↑ 9.2% | $372.60 \pm 28.52$ | $418.10 \pm 13.90$ | $+45.50$ | ↑ 12.2% |
| Reacher-v4 | $-5.98 \pm 0.27$ | $-4.88 \pm 0.49$ | $+1.10$ | ↑ 18.4% | $-5.88 \pm 0.44$ | $-5.17 \pm 0.32$ | $+0.71$ | ↑ 12.1% |
| Average | $1522.17 \pm 99.62$ | $1745.43 \pm 86.85$ | $+223.27$ | ↑ 14.7% | $1501.81 \pm 102.80$ | $2105.56 \pm 111.82$ | $+603.75$ | ↑ 40.2% |

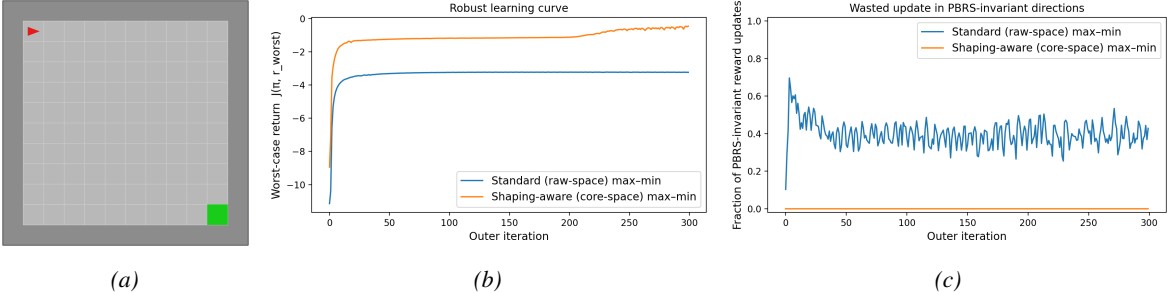

*(a)*          *(b)*          *(c)*

*Figure 2.* MiniGrid evaluation comparing the standard max-min baseline and our shaping-aware method. (a) Grid-world task: the agent starts in the top-left cell and must reach the goal in the bottom-right cell. (b) Worst-case return as a function of training iteration. (c) Fraction of each adversary update that lies in PBRS-invariant directions.

sense that it can meaningfully affect the worst-case objective. Together, these results support our motivating claim: removing PBRS redundancy makes adversary updates behaviorally identifiable, which translates into more effective robust training and better final worst-case performance.

**Comparison on MuJoCo tasks.** To evaluate how well each method performs under the nominal environments, we report the results in Table 1. Across all five MuJoCo tasks, our shaping-aware variant consistently outperforms the corresponding robust RL baseline. Under the entropy-based objective, the gains range from 9.2% on Humanoid to 18.4% on Reacher (average a 14.7% improvement). Under the KL-based objective, the improvements are even larger, with particularly strong gains on HalfCheetah (107.7%) and Walker2d (33.6%), yielding an average improvement of 35.7%. Overall, these results show that removing PBRS-induced redundancy via our shaping-aware projection consistently improves robust policy learning and yields higher worst-case returns across environments.

Figure 3 compares learning curves for both methods across five MuJoCo tasks to solve the KL-based problem and we leave the result for entropy-based problem in Appendix D.2. Overall, our method achieves higher returns throughout training and typically exhibits more stable optimization dynamics. The improvement is especially clear on HalfCheetah, where the shaping-aware policy stays above the baseline at essentially every timestep and continues to improve late in training, while the baseline progresses more slowly and plateaus earlier. Beyond final performance, the curves also show a consistent sample-efficiency advantage. On Reacher,

our method reaches near-solved performance almost immediately, while the baseline requires substantially more interaction (on the order of $2 \times 10^5$ timesteps) to catch up. On Hopper and Humanoid, the two methods share similar early-stage trends, but our method converges to a better asymptotic return and tends to reduce training instability as learning proceeds. Taken together, these results indicate that the shaping-aware projection not only improves final performance, but also enables the agent to reach a given performance level with fewer environment interactions, i.e., it converges faster than the standard reward-robust baseline.

**Testing for reward robustness.** To assess robustness to reward uncertainty, we evaluate all methods under structured reward perturbations that change the task incentives. For the locomotion tasks (HalfCheetah, Hopper, Walker2d, and Humanoid), the nominal reward primarily encourages forward progress, and moving backward typically yields a non-positive reward. Following the protocol of (Husain et al., 2021), during training, we flip this incentive by adding random reward shifts, so that backward motion can also be beneficial under the perturbed rewards. We report performance under the original reward during evaluation. For Reacher, where the objective is to reach a target position, we adopt the evaluation protocol of (Ashlag et al., 2025). All policies are trained on the nominal environment. At test time, we relocate the target to a random position at varying distances from the nominal location and evaluate performance with respect to the reward induced by this relocated target. Implementation details for constructing the perturbed rewards are provided in Appendix C.3. Overall,

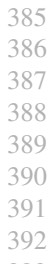
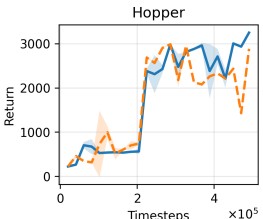
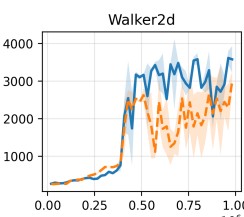
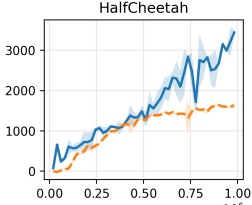
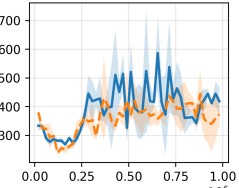
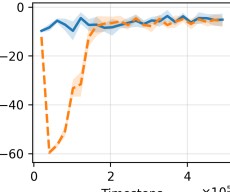

*Figure 3.* Mean and standard deviation return of the evaluation policy over 10 rollouts and 5 seeds for KL-based reward-robust problem, reported every 20k environment steps. The return is in terms of the environment's original reward.

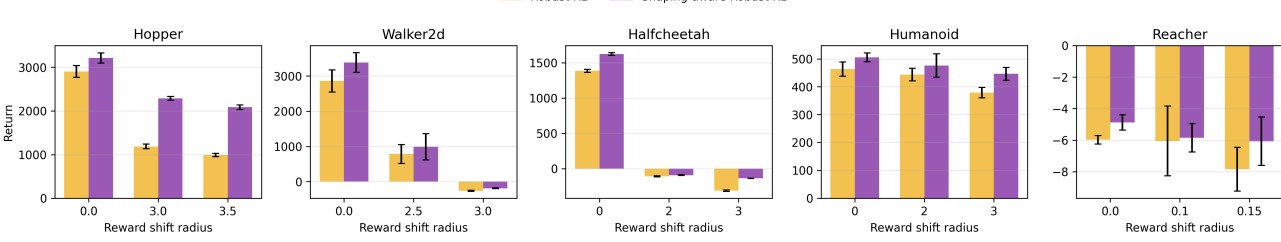

*Figure 4.* Mean $\pm$ 95% confidence interval return of the evaluation policy under reward perturbation over 100 episodes and 5 seeds for the entropy-based problem.

these perturbations introduce systematic and interpretable reward misspecification across domains, allowing us to compare how well each method maintains performance when the reward deviates from the nominal specification.

Figure 8 reports robustness under reward perturbations for the entropy-based problem, and we include the results for KL-based problem in Appendix D.3. Overall, the shaping-aware method exhibits smaller performance degradation than the robust RL baseline as the reward shift radius increases, indicating improved stability to reward misspecification. The perturbations affect HalfCheetah and Walker2d most strongly, where returns drop sharply for both methods as the shift radius grows, suggesting these locomotion tasks are particularly sensitive to reward changes. In contrast, Hopper, Humanoid, and Reacher show more moderate changes with increasing shift radius, and the shaping-aware method consistently maintains higher returns than the baseline across perturbation levels.

*Table 2.* Ablation on canonicalization choices under reward perturbations for Walker2d with the entropy-based robust objective (reward shift radius = 3). Results are reported as mean $\pm$ 95% confidence interval over 5 random seeds.

| Method | Walker2d |
|---|---|
| Reward-Robust (baseline) | -259.59$\pm$16.07 |
| Shaping-aware (ours, $\Omega^*$-projection) | -188.14$\pm$9.26 |
| Shaping-aware (random $\Phi$ canonicalization) | -226.17$\pm$36.72 |
| Shaping-aware ($L^2$ projection) | -199.04$\pm$9.37 |

**Ablation on projection choices.** To isolate the contribution of our $\Omega^*$-aligned canonicalization, Table 2 compares

several projection choices under reward perturbations. Implementation details are provided in Appendix C.4, and full results for all environments are reported in Appendix D.5. First, we include a random $\Phi$ variant, which injects an arbitrary PBRS shaping term into the reward. This variant does not yield meaningful robustness improvements, indicating that the gains of our method do not come from simply adding an extra shaping-like component, but from learning a canonicalization that removes PBRS redundancy in a principled way. We also consider an $L^2$ projection that performs orthogonal projection regardless of the uncertainty-set geometry. This variant still removes PBRS-equivalent directions and thus improves robustness relative to the baseline, confirming that redundancy removal alone is beneficial. However, the $\Omega^*$-aligned projection performs better, suggesting that matching the canonicalization to the robustness geometry provides additional gains under reward perturbations.

# 6. Conclusion

In this paper, we propose **Shaping-Aware Reward-Robust RL**, which constructs behavior discriminative uncertainty sets over PBRS equivalence classes and enables practical training via a projection-learning module compatible with standard RL optimizers. Across a discrete grid-world benchmark and five MuJoCo tasks, our method consistently improves robust performance and convergence over representative baselines. These results support our central claim that eliminating behaviorally equivalent rewards leads to more effective robust policy learning under reward uncertainty.

## Impact Statement

This paper presents work whose goal is to advance the field of Machine Learning. There are many potential societal consequences of our work, none which we feel must be specifically highlighted here.

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

# A. Related Work

## A.1. Reward-Robust RL and Regularization RL

Reward-robust RL studies how to learn policies that remain effective when the reward signal is perturbed in an adversarial manner (Mannor et al., 2007), and has been explored in a broad range of settings (Morimoto & Doya, 2005; Wang et al., 2020; He & Lv, 2023; Yan et al., 2024). A common view is to think of the problem as a dynamic zero-sum game: the agent selects a policy to maximize return, while an adversary selects a reward within a prescribed uncertainty set to minimize it (Iyengar, 2005; Wiesemann et al., 2013). This leads to a max-min optimization problem that is often expensive to solve, creating practical bottlenecks for large-scale applications (Tessler et al., 2019). To improve tractability, recent work has pursued several directions. Some approaches exploit structured uncertainty sets that admit more efficient optimization (Mannor et al., 2016; 2012; Goyal & Grand-Clement, 2023). In contrast, a larger body of work adopts rectangular uncertainty across states or state-action pairs to derive scalable solvers (Behzadian et al., 2021; Ho et al., 2021; Bagnell et al., 2001; Grand-Clément & Kroer, 2021). In parallel, the strong empirical performance of regularized policy optimization has motivated many methods that use explicit regularizers to encourage desirable behavior, such as better exploration (Haarnoja et al., 2017; Lee et al., 2018; Ahmed et al., 2019; Neu et al., 2017), improved training stability (Schulman et al., 2015; Haarnoja et al., 2018), and improved sample efficiency (Seo et al., 2021).

More recently, there is growing recognition that certain robust RL objectives are closely connected to regularized RL formulations. Such robustness-regularization duality typically relies on the Legendre-Fenchel transform, which has been well established in statistical learning (Xu et al., 2009; Hoffman et al., 2019; Duchi et al., 2021; Shafieezadeh Abadeh et al., 2015). In the RL setting, (Derman et al., 2023) formally proves that policy regularization methods solve a particular instance of robust MDPs with uncertain rewards. A representative example is (Eysenbach & Levine, 2021), which shows that maximum policy entropy regularized RL maximizes a lower bound on a reward-robust RL objective, and thus can be used to learn policies that are robust to some disturbances in the reward function. Subsequent work has expanded this perspective beyond policy regularization to state- and state-action-based regularizer, which induce robustness against adversaries that depend on global patterns rather than only local information (Islam et al., 2019; Seo et al., 2021; Yuan et al., 2022; Kim et al., 2023; Bolland et al., 2024). In particular, (Husain et al., 2021) studies a broad family of regularized RL objectives and derives the corresponding adversarial-reward dual problems, and (Ashlag et al., 2025) analyzes the special case of state-entropy regularization, demonstrating improved robustness to structured, spatially correlated perturbations.

In this work, we build on the same Legendre-Fenchel duality and show that the regularization-robustness correspondence can be extended to our shaping-aware reward-robust objective. This equivalence enables us to avoid solving the inner minimization on the projected uncertainty set directly and instead optimize an associated regularized objective, substantially reducing the computational overhead of our approach.

## A.2. Potential-Based Reward Shaping

Potential-based reward shaping (PBRS) was introduced by (Ng et al., 1999) as a principled way to modify rewards without changing the optimal policy. The key idea is to add a shaping term defined by the difference of a potential function evaluated at two states, which can be canceled out in the Bellman equation, thus preserving the optimal policy (Wiewiora, 2003; Asmuth et al., 2008). PBRS is generally used to incorporate heuristics or task-agnostic metrics that encourage broader state exploration (Ma et al., 2024). Accordingly, much of the literature focuses on how to construct effective potential functions. In some large-scale practical RL tasks, the potential functions are heavily handcrafted based on prior knowledge (Berner et al., 2019; Ye et al., 2020; Yu et al., 2020). Besides reward engineering, a distinct line of work applies exploration bonuses to reward novel or infrequently visited states (Mahankali et al., 2024; Liu et al., 2025a; Devidze et al., 2022; Badia et al., 2020). For example, count-based reward shaping tracks visitation counts and assigns higher rewards to less frequently visited states (Lobel et al., 2023; Machado et al., 2020; Choshen et al., 2018). However, in continuous spaces, exact state counting is typically infeasible, and prior work mainly relies on approximations. For example, (Tang et al., 2017) uses hashing to discretize the state space and enable count-style bonuses, while (Bellemare et al., 2016) proposes pseudo-counts derived from recording probabilities. Beyond count-based reward shaping, several works incorporate structured domain knowledge into shaping to guide exploration more directly, for example, by using heuristic shaping to reduce the effective horizon (Cheng et al., 2021) or by exploiting shaping to narrow the portion of the state space that must be searched (Gupta et al., 2022). Recent approaches also consider alternative feedback signals, such as success rates derived from historical experience as the shaped rewards, to balance exploration and exploitation in a more adaptive way (Ma et al., 2024).

Beyond exploration, PBRS has also been used as a general mechanism for integrating task structure and prior knowledge

into RL objectives. For instance, (Gao et al., 2024) proposes a reachability-based shaping signal for hierarchical RL to alleviate the negative interference of suboptimal demonstrations, where shaped rewards encode progress toward subgoals, enabling efficient skill learning. In addition, reward shaping has been adopted in deep RL practice as a way to mitigate sparse reward problems. (Lu et al., 2019) introduces a predictive coding-based shaping objective that understands the structure and dynamics of the environment and emphasize on features most useful for learning, improving learning stability under sparse rewards. More recently, reward shaping has been revisited through the lens of foundation models. Text2Reward (Xie et al., 2023) uses language models to generate reward signals from natural-language task descriptions, reflecting an emerging direction where shaping is produced from semantic feedback rather than hand-designed potentials.

Our paper is also related to a broader line of work on reward design and reward optimization as a method of shaping agent behavior. For instance, behavior alignment via reward optimization directly treats the reward as a design variable to induce desired behaviors (Gupta et al., 2023), while online reward selection methods intertwine reward selection with policy optimization to improve sample efficiency in reward design loops (Zhang et al., 2024). In RLHF, recent work studies how to improve the discriminative power of learned reward models (e.g., via contrastive objectives), which are conceptually adjacent to our emphasis on behavior-discriminative reward representations (Chen et al., 2024).

In contrast to prior work, which primarily designs shaped rewards to improve exploration or induce desired behaviors, we use reward shaping as a tool for robustness modeling. To the best of our knowledge, we are the first to bring PBRS into reward-robust RL to construct uncertainty sets over behaviorally meaningful variations rather than variations in raw reward values. Leveraging PBRS's policy invariance property, we identify and remove shaping-induced redundancies in standard uncertainty sets, yielding a reduced set in which each remaining reward corresponds to a behaviorally distinct objective.

## B. Proofs

### B.1. Proof of Lemma 3.1

*Proof.* Fix any policy $\pi$ and initial state $s_0 = s$. For all $s \in \mathcal{S}$, define $v_r^\pi(s) = \mathbb{E}\left[\sum_{t=0}^\infty \gamma^t r(s_t, a_t, s_{t+1}) \,\middle|\, s_0 = s, \pi, P\right]$ as the value function at state $s$. Consider the discounted return under reward $r'$:

$$v_{r'}^\pi(s) = \mathbb{E}\left[\sum_{t=0}^\infty \gamma^t \big(r(s_t, a_t, s_{t+1}) + (\gamma\Phi(s_{t+1}) - \Phi(s_t))\big) \,\middle|\, s_0 = s, \pi, P\right]$$

$$= v_r^\pi(s) \;+\; \mathbb{E}\left[\sum_{t=0}^\infty \gamma^{t+1}\Phi(s_{t+1}) - \sum_{t=0}^\infty \gamma^t \Phi(s_t) \,\middle|\, s_0 = s, \pi, P\right].$$

The first sum can be re-indexed as $\sum_{t=0}^\infty \gamma^{t+1}\Phi(s_{t+1}) = \sum_{t=1}^\infty \gamma^t \Phi(s_t)$. Therefore the two series telescope:

$$\sum_{t=1}^\infty \gamma^t \Phi(s_t) - \sum_{t=0}^\infty \gamma^t \Phi(s_t) = -\Phi(s_0).$$

Hence

$$v_{r'}^\pi(s) = v_r^\pi(s) - \Phi(s_0),$$

and consequently

$$J(\pi, r') = \mathbb{E}_{s_0 \sim \mu_0}\big[v_r^\pi(s_0)\big] = J(\pi, r) - \mathbb{E}_{s_0 \sim \mu_0}[\Phi(s_0)].$$

The shift $-\mathbb{E}_{s_0 \sim \mu_0}[\Phi(s_0)]$ does not depend on $\pi$. So within each equivalence class, all rewards differ by a policy-independent constant shift. This completes the proof. $\square$

### B.2. Proof of Proposition 3.2

*Proof.* Fix $(\mu_0, \pi, P)$ and view rewards as vectors in the finite-dimensional space $\mathbb{R}^{\mathcal{X}'}$. Since $\mathcal{U}$ has nonempty interior around $\tilde{r}$, for any chosen norm $\|\cdot\|$ on $\mathbb{R}^{\mathcal{X}'}$, there exists $\rho > 0$ such that the open ball

$$B(\tilde{r}, \rho) \;:=\; \{u \in \mathbb{R}^{\mathcal{X}'} : \|u - \tilde{r}\| < \rho\} \subseteq \mathcal{U}.$$

Next, choose any potential function $\Phi \in \mathbb{R}^{\mathcal{S}}$ that is not constant (e.g., pick two states $s_1 \neq s_2$ and set $\Phi(s_1) = 1, \Phi(s_2) = 0$, and $\Phi = 0$ elsewhere). Define the corresponding shaping term $F \in \mathbb{R}^{\mathcal{X}'}$ by $F(s, a, s') = \gamma\Phi(s') - \Phi(s)$. Because $\Phi$ is not

constant, $F \neq 0$, hence $\|F\| > 0$. Let $\delta := \rho/\|F\|$. Then for any $\alpha \in \mathbb{R}$ with $|\alpha| \leq \delta$,

$$\|\tilde{r} + \alpha F - \tilde{r}\| = \|\alpha F\| \leq |\alpha| \, \|F\| \leq \delta \|F\| = \rho,$$

which implies $\tilde{r} + \alpha F \in B(\tilde{r}, \rho) \subseteq \mathcal{U}$.

It remains to show PBRS equivalence. Fix any policy $\pi$ and initial state $s_0 = s$. For all $s \in \mathcal{S}$, define $v_r^\pi(s) = \mathbb{E}\left[\sum_{t=0}^{\infty} \gamma^t r(s_t, a_t, s_{t+1}) \mid s_0 = s, \pi, P\right]$ as the value function at state $s$. Consider the discounted return under reward $\tilde{r} + \alpha F$:

$$v_{\tilde{r}+\alpha F}^\pi(s) = \mathbb{E}\left[\sum_{t=0}^{\infty} \gamma^t \big(\tilde{r}(s_t, a_t, s_{t+1}) + \alpha(\gamma \Phi(s_{t+1}) - \Phi(s_t))\big) \,\Big|\, s_0 = s, \pi, P\right]$$

$$= v_{\tilde{r}}^\pi(s) \; + \; \alpha \, \mathbb{E}\left[\sum_{t=0}^{\infty} \gamma^{t+1} \Phi(s_{t+1}) - \sum_{t=0}^{\infty} \gamma^t \Phi(s_t) \,\Big|\, s_0 = s, \pi, P\right].$$

The first sum can be re-indexed as $\sum_{t=0}^{\infty} \gamma^{t+1} \Phi(s_{t+1}) = \sum_{t=1}^{\infty} \gamma^t \Phi(s_t)$. Therefore the two series telescope:

$$\sum_{t=1}^{\infty} \gamma^t \Phi(s_t) - \sum_{t=0}^{\infty} \gamma^t \Phi(s_t) = -\Phi(s_0).$$

Hence

$$v_{\tilde{r}+\alpha F}^\pi(s) = v_{\tilde{r}}^\pi(s) - \alpha \, \Phi(s_0),$$

and consequently

$$J(\pi, \tilde{r} + \alpha F) = \mathbb{E}_{s_0 \sim \mu_0}\big[v_{\tilde{r}+\alpha F}^\pi(s_0)\big] = J(\pi, \tilde{r}) - \alpha \, \mathbb{E}_{s_0 \sim \mu_0}[\Phi(s_0)].$$

The shift $-\alpha \, \mathbb{E}_{s_0 \sim \mu_0}[\Phi(s_0)]$ does not depend on $\pi$, so for any two policies $\pi_1, \pi_2$,

$$J(\pi_1, \tilde{r} + \alpha F) - J(\pi_2, \tilde{r} + \alpha F) = J(\pi_1, \tilde{r}) - J(\pi_2, \tilde{r}).$$

Thus all rewards in $\{\tilde{r} + \alpha F : |\alpha| \leq \delta\}$ induce identical policy rankings, i.e., they are PBRS-equivalent. This completes the proof. $\qquad\square$

### B.3. Proof of Theorem 3.3

*Proof.* Take any $\tilde{r} \in \mathcal{R}$. By construction of the projection Proj, we have $\text{Proj}(\tilde{r}) = \tilde{r} + F_\Phi$ for some shaping term $F_\Phi \in \mathcal{S}_{\text{shape}}$. By Lemma 3.1, we have

$$J(\pi, \text{Proj}(\tilde{r})) = J(\pi, \tilde{r} + F_\Phi) = J(\pi, \tilde{r}) \qquad \forall \, \pi. \qquad (4)$$

So for each reward, its value is exactly the same as its projected version.

Let

$$m := \min_{\tilde{r} \in \mathcal{U}} J(\pi, \tilde{r}), \qquad m_{\text{core}} := \min_{\tilde{r}_{\text{core}} \in \mathcal{U}_{\text{core}}} J(\pi, \tilde{r}_{\text{core}}).$$

Take any $\tilde{r} \in \mathcal{U}$. From Equation 4, we have

$$J(\pi, \tilde{r}) = J(\pi, \text{Proj}(\tilde{r})) \geq m_{\text{core}},$$

because $\text{Proj}(\tilde{r}) \in \mathcal{U}_{\text{core}}$ by definition and $m_{\text{core}}$ is the minimum over that set.

Since this holds for every $\tilde{r} \in \mathcal{U}$, we can take the minimum over $\tilde{r} \in \mathcal{U}$ on the left:

$$m = \min_{\tilde{r} \in \mathcal{U}} J(\pi, \tilde{r}) \geq m_{\text{core}}. \qquad (5)$$

Now take any $\tilde{r}_{\text{core}} \in \mathcal{U}_{\text{core}}$. By definition of $\mathcal{U}_{\text{core}}$, there exists some $\tilde{r} \in \mathcal{U}$ such that

$$\tilde{r}_{\text{core}} = \text{Proj}(\tilde{r}).$$

Using Lemma 3.1 again,
$$J(\pi, \tilde{r}_{\text{core}}) = J(\pi, \tilde{r}).$$

By definition of $m$,
$$m := \min_{\tilde{r} \in \mathcal{U}} J(\pi, \tilde{r}) \ \leq \ J(\pi, \tilde{r}) = J(\pi, \tilde{r}_{\text{core}}).$$

Since this holds for every $\tilde{r}_{\text{core}} \in \mathcal{U}_{\text{core}}$, we can take the minimum over $\tilde{r}_{\text{core}} \in \mathcal{U}_{\text{core}}$ on the right-hand side:
$$m \ \leq \ \min_{\tilde{r}_{\text{core}} \in \mathcal{U}_{\text{core}}} J(\pi, \tilde{r}_{\text{core}}) = m_{\text{core}}. \tag{6}$$

From Equation 5 and Equation 6, we have
$$m \geq m_{\text{core}} \quad \text{and} \quad m \leq m_{\text{core}} \ \Rightarrow \ m = m_{\text{core}}.$$

Thus, for any policy $\pi$,
$$\min_{\tilde{r} \in \mathcal{U}} J(\pi, \tilde{r}) = \min_{\tilde{r}_{\text{core}} \in \mathcal{U}_{\text{core}}} J(\pi, \tilde{r}_{\text{core}}).$$

Now take the max over policies:
$$\max_\pi \min_{\tilde{r} \in \mathcal{U}} J(\pi, \tilde{r}) = \max_\pi \min_{\tilde{r}_{\text{core}} \in \mathcal{U}_{\text{core}}} J(\pi, \tilde{r}_{\text{core}}).$$

Because the inner quantities are identical for each $\pi$, the set of maximizers $\arg\max_\pi$ is also the same for the original and projected uncertainty sets. This shows that the robust optimal value is unchanged when we pass from $\mathcal{U}$ to $\mathcal{U}_{\text{core}}$.

$\square$

### B.4. Proof of Proposition 3.4

*Proof.* Let $A := \text{aff}(\Theta)$ be the affine hull of $\Theta$, and let $k := \dim(A) = \dim(\Theta)$, the affine dimension of $\Theta$. Pick any fixed $\theta_0 \in \Theta$, and define the associated linear subspace
$$V := A - \theta_0 = \{\theta - \theta_0 : \theta \in A\} \subseteq \mathbb{R}^d.$$

Then $V$ is a $k$-dimensional linear subspace, i.e. $\dim(V) = k$. Similarly, define the intersection of the shaping subspace with this affine subspace: $S' := \mathcal{S}_{\text{shape}} \cap A$. Then the corresponding linear subspace inside $V$ is
$$S := S' - \theta_0 = \mathcal{S}_{\text{shape}} \cap V.$$

where $\dim(S) = \dim(\mathcal{S}_{\text{shape}} \cap A)$. Define a shifted version of $\text{Proj}$ on $V$:
$$T : V \to \mathbb{R}^d, \qquad T(v) := \text{Proj}(\theta_0 + v) - \text{Proj}(\theta_0).$$

Intuitively, $T$ describes how $\text{Proj}$ moves points relative to the base point $\theta_0$. Observe that, for any $v \in V$ and any $u \in S$ (so $u \in \mathcal{S}_{\text{shape}} \cap V$), we have $\theta_0 + v + u \in A$. This is because $u \in V$ by definition of $u$ and $V$ is a linear subspace, so $v + u \in V$. Thus, we have $\theta_0 + (v + u) \in \theta_0 + V = A$ by definition of $V$. Next, by shaping invariance, we have $\text{Proj}(\theta_0 + v + u) = \text{Proj}(\theta_0 + v)$. Therefore,
$$T(v + u) = \text{Proj}(\theta_0 + v + u) - \text{Proj}(\theta_0) = \text{Proj}(\theta_0 + v) - \text{Proj}(\theta_0) = T(v).$$

So $T$ is constant along directions in $S$. That means points that differ by something in $S$ are indistinguishable to $T$. In other words, $T$ factors through the quotient space $V/S$. Let $C : V \to V/S, C(v) := [v]$ be the canonical quotient map sending each $v \in V$ to its equivalence class $[v]$. Because $T(v + u) = T(v)$ for all $u \in S$, if two vectors are equivalent ($v' \sim v$), then they get mapped to the same output: $v' = v + u, \ u \in S \implies T(v') = T(v)$. So $T$ only depends on the equivalence class $[v]$, not the specific representative. We define:
$$\tilde{T} : V/S \to \mathbb{R}^d, \qquad \tilde{T}([v]) := T(v).$$

And we can write:
$$T = \tilde{T} \circ C,$$
because for any $v \in V$,
$$(\tilde{T} \circ C)(v) = \tilde{T}([v]) = T(v).$$
By construction, $\tilde{T}$ is linear, and its image is the linear span of $T(V)$:
$$\operatorname{Im}(\tilde{T}) = \operatorname{span}(T(V)).$$

The dimension of the quotient space is
$$\dim(V/S) = \dim(V) - \dim(S) = k - \dim(\mathcal{S}_{\text{shape}} \cap A).$$

Since $\tilde{T}$ is a linear map from $V/S$ to $\mathbb{R}^d$, its rank is at most the dimension of its domain:
$$\operatorname{rank}(\tilde{T}) \leq \dim(V/S) = k - \dim(\mathcal{S}_{\text{shape}} \cap A).$$

And notice that $\operatorname{rank}(\tilde{T}) = \dim(\operatorname{Im}(\tilde{T})) = \dim(\operatorname{span}(T(V)))$. By definition, $\Theta_{\text{core}} = \{\operatorname{Proj}(\theta) : \theta \in \Theta\}$. Any $\theta \in \Theta$ can be written as $\theta = \theta_0 + v$ with $v \in V$. Then
$$\operatorname{Proj}(\theta) = \operatorname{Proj}(\theta_0 + v) = \operatorname{Proj}(\theta_0) + T(v).$$

Therefore
$$\Theta_{\text{core}} \subseteq \operatorname{Proj}(\theta_0) + T(V),$$
and consequently,
$$\operatorname{aff}(\Theta_{\text{core}}) \subseteq \operatorname{Proj}(\theta_0) + \operatorname{span}(T(V)).$$

Thus the affine dimension of $\Theta_{\text{core}}$ satisfies
$$\dim(\Theta_{\text{core}}) = \dim\big(\operatorname{aff}(\Theta_{\text{core}})\big) \leq \dim\big(\operatorname{span}(T(V))\big) = \operatorname{rank}(\tilde{T}) \leq k - \dim(\mathcal{S}_{\text{shape}} \cap A).$$

Recall $k = \dim(A) = \dim(\Theta)$, so we obtain:
$$\dim(\Theta_{\text{core}}) \leq \dim(\Theta) - \dim\big(\mathcal{S}_{\text{shape}} \cap \operatorname{aff}(\Theta)\big).$$
which completes the proof. If $\mathcal{S}_{\text{shape}} \cap \operatorname{aff}(\Theta)$ is nontrivial (i.e., has positive dimension), then
$$\dim\big(\mathcal{S}_{\text{shape}} \cap \operatorname{aff}(\Theta)\big) > 0,$$
so
$$\dim(\Theta_{\text{core}}) \leq \dim(\Theta) - \underbrace{\dim\big(\mathcal{S}_{\text{shape}} \cap \operatorname{aff}(\Theta)\big)}_{>0} < \dim(\Theta),$$

i.e., the affine dimension of the projected ambiguity set is strictly smaller than that of the original ambiguity set. $\square$

### B.5. Dimension Drop for Multi-Armed Bandits

**Proposition B.1.** *Consider a discounted multi-armed bandit with a single state $s$ and action set $\mathcal{A}$. Let $\mathcal{R} = \mathbb{R}^{\mathcal{A}}$ denote the reward space, and let $\mathcal{S}_{\text{shape}}$ be the shaping subspace. Then the quotient space $\mathcal{R}_{\text{core}} = \mathcal{R}/\mathcal{S}_{\text{shape}}$ has dimension $|\mathcal{A}| - 1$. Equivalently, projecting to canonical representatives removes exactly one behaviorally redundant direction.*

*Proof.* PBRS shaping is
$$F(s, a, s') = \gamma \Phi(s') - \Phi(s).$$

For multi-armed bandit MPD, there is only one state, so $s' = s$, hence the PBRS shaping term

$$F(s, a, s) \;=\; \gamma\Phi(s) - \Phi(s) \;=\; (\gamma - 1)\Phi(s) \;=:\; F$$

is a constant independent of the action. Thus, shaping can only add the same constant $F$ to every reward. Therefore, two reward vectors $r, r' \in \mathbb{R}^{|\mathcal{A}|}$ are PBRS-equivalent iff

$$r' \;=\; r + F \cdot \mathbf{1},$$

i.e., they differ by the all-ones direction. Then the shaping subspace is

$$\mathcal{S}_{\text{shape}} \;=\; \{F \cdot \mathbf{1} : F \in \mathbb{R}\} \;=\; \text{span}\{\mathbf{1}\},$$

which has dimension 1. Therefore, the quotient space has dimension

$$\dim\big(\mathcal{R}/\mathcal{S}_{\text{shape}}\big) \;=\; |\mathcal{A}| - 1.$$

This completes the proof. $\qquad\square$

### B.6. Additional Properties of Shaping-aware Projected Uncertainty Set

In this section, we propose additional properties of the projected uncertainty set.

**Proposition B.2.** *Let $\mathcal{U} \subseteq \mathcal{R}$ be a compact uncertainty set and $\mathcal{U}_{core} := \text{Proj}(\mathcal{U}) = \{\text{Proj}(r) : r \in \mathcal{U}\}$ is the corresponding projected uncertainty set. If $\text{Proj} : \mathcal{R} \to \mathcal{R}$ is continuous, then $\mathcal{U}_{core}$ is compact.*

*Proof.* Since $\text{Proj}$ is continuous, the continuous image of a compact set is compact. Hence $\mathcal{U}_{core}$ is compact. $\qquad\square$

**Proposition B.3.** *Let $\mathcal{U} \subseteq \mathcal{R}$ be a convex uncertainty set and $\mathcal{U}_{core} := \text{Proj}(\mathcal{U}) = \{\text{Proj}(r) : r \in \mathcal{U}\}$ is the corresponding projected uncertainty set. If $\text{Proj} : \mathcal{R} \to \mathcal{R}$ is affine, i.e., for all $r_1, r_2 \in \mathcal{R}$ and $\lambda \in [0, 1]$,*

$$\text{Proj}((1 - \lambda)r_1 + \lambda r_2) = (1 - \lambda)\text{Proj}(r_1) + \lambda\text{Proj}(r_2),$$

*then $\mathcal{U}_{core}$ is convex.*

*Proof.* Take any $u'_1, u'_2 \in \mathcal{U}_{\text{core}}$. Then there exist $u_1, u_2 \in \mathcal{U}$ such that $u'_i = \text{Proj}(u_i)$ for $i \in \{1, 2\}$. For any $\lambda \in [0, 1]$, since $\mathcal{U}$ is convex, we have $(1 - \lambda)u_1 + \lambda u_2 \in \mathcal{U}$. If $\text{Proj}$ is affine, then we have

$$(1 - \lambda)u'_1 + \lambda u'_2 = (1 - \lambda)\text{Proj}(u_1) + \lambda\text{Proj}(u_2) = \text{Proj}((1 - \lambda)u_1 + \lambda u_2) \in \mathcal{U}_{\text{core}}.$$

Thus $\mathcal{U}_{\text{core}}$ is convex. $\qquad\square$

### B.7. Proof of Theorem 4.1

To prove Theorem 4.1, we first recall a standard connection between regularization and robustness with penalty. The statement below is adapted from Theorem 1 of (Husain et al., 2021) to match our notation.

**Theorem B.4.** *Let $\Omega$ be proper, closed, and strongly convex, and let $\Omega^*$ denote its convex conjugate. Then, for any policy $\pi$,*

$$J(\pi, r) - \Omega(\pi) = \min_{\tilde{r} \in \mathbb{R}^{\mathcal{X}'}} \Big\{ J(\pi, \tilde{r}) + \Omega^*(r - \tilde{r}) \Big\}.$$

*Proof.* (Sketch) The key part of the proof is to rewrite $\Omega$ in terms of the convex conjugate of $-\Omega$, which is well-defined since we assume $\Omega$ to be strongly convex. The detailed proof can be found in (Husain et al., 2021). $\qquad\square$

We now turn to budgeted uncertainty sets. For $\epsilon \geq 0$ and a regularization weight $\lambda > 0$, define

$$\mathcal{U} := \Big\{ \tilde{r} \in \mathbb{R}^{\mathcal{X}'} \,:\, \Omega^*\Big(\frac{r - \tilde{r}}{\lambda}\Big) \leq \epsilon \Big\}.$$

Next, we assume the Slater condition holds:

**Assumption B.5** (Slater condition). There exists $\bar{r} \in \mathbb{R}^{\mathcal{X}'}$ such that $\Omega^*(r/\lambda - \bar{r}/\lambda) < \epsilon$.

The Slater condition requires that $\mathcal{U}$ contain at least one strictly feasible solution, which is a standard assumption ensuring strong duality for the minimization over $\tilde{r}$. When needed, this condition can be enforced by a mild relaxation of the budget $\epsilon$, so we treat it as a standing assumption. Under this assumption, we can prove an equivalence between reward-robust RL and an appropriately regularized RL objective, as shown next.

**Theorem B.6.** *Fix any policy $\pi$ and $\epsilon \geq 0$. Under Assumption B.5, the following duality holds:*

$$\min_{\tilde{r} \in \mathcal{U}} J(\pi, \tilde{r}) = J(\pi, r) - \lambda(\Omega(\pi) - \epsilon),$$

*Proof.* Consider the inner minimization problem

$$\min_{\tilde{r}} \ J(\pi, \tilde{r}) \quad \text{s.t.} \quad \Omega^*\left(\frac{r - \tilde{r}}{\lambda}\right) \leq \epsilon.$$

For fixed $\pi$, the objective is linear in $\tilde{r}$ and the constraint is convex in $\tilde{r}$ because $\Omega^*$ is convex. Under Slater's condition, strong duality holds, hence

$$\min_{\tilde{r} \in \mathcal{U}} J(\pi, \tilde{r}) = \max_{\lambda \geq 0} \min_{\tilde{r}} \left( J(\pi, \tilde{r}) + \lambda\left(\Omega^*\left(\frac{r - \tilde{r}}{\lambda}\right) - \epsilon\right)\right),$$

and in particular there exists an optimal multiplier $\lambda^\star \geq 0$ attaining the maximum.

Fix such $\lambda^\star$ and we will continue use $\lambda$ instead of $\lambda^*$ for convenience, which is also adopted by (Eysenbach & Levine, 2021; Ashlag et al., 2025). Recall that the convex conjugate of $\lambda \Omega(\cdot)$ is $\lambda \Omega^*(\frac{\cdot}{\lambda})$, applying Theorem B.4 with the regularizer $\lambda\Omega$ yields

$$\min_{\tilde{r} \in \mathcal{U}} J(\pi, \tilde{r}) = \min_{\tilde{r}}\left\{ J(\pi, \tilde{r}) + \lambda(\Omega^*\left(\frac{r - \tilde{r}}{\lambda}\right) - \epsilon)\right\} = J(\pi, r) - \lambda(\Omega(\pi) - \epsilon).$$

This completes the proof. $\square$

Theorem 4.1 holds because the equivalence above is agnostic to the particular reward domain. It only requires that the inner problem optimizes over an uncertainty set defined through $\Omega^*$ and that Slater's condition holds. In our setting, the projection $\mathrm{Proj}$ maps each reward to a PBRS-equivalent canonical representative and satisfies $\mathrm{Proj}(r + F_\Phi) = \mathrm{Proj}(r)$. Therefore, replacing the original uncertainty set by its projected counterpart $\mathcal{U}_{\text{core}} = \mathrm{Proj}(\mathcal{U})$ simply restricts the adversary to one representative per PBRS class without altering policy rankings. The same duality applies verbatim on $\mathcal{U}_{\text{core}}$. Taking the maximum over $\pi$ and removing the $\lambda\epsilon$ term (as it does not affect policy optimization) completes the proof.

## B.8. Properties of $\Omega^*$-induced Projection

Recall that we estimate the projection using

$$\mathrm{Proj}(r) = \arg\min_{\Phi} \Omega^*\Big(r(s, a, s') - \big(\gamma\Phi(s') - \Phi(s)\big)\Big).$$

Next, we establish conditions under which the projection is well-defined, i.e., the minimizer exists and is unique. We begin with a technical lemma that will be used in the proof.

**Lemma B.7.** $\mathcal{S}_{shape}$ *is a linear subspce.*

*Proof.* Recall that PBRS shaping terms are of the form

$$F_\Phi(s, a, s') \ = \ \gamma\Phi(s') - \Phi(s).$$

Then we define the shaping set as

$$S_{\text{shape}} := \{F_\Phi : \Phi \in \mathbb{R}^{\mathcal{S}}\}.$$

Now take any $\Phi_1, \Phi_2$ and scalars $a, b$. Because the map $\Phi \mapsto F_\Phi$ is linear,

$$F_{a\Phi_1 + b\Phi_2}(s, a, s') = \gamma(a\Phi_1 + b\Phi_2)(s') - (a\Phi_1 + b\Phi_2)(s) = aF_{\Phi_1}(s, a, s') + bF_{\Phi_2}(s, a, s').$$

So:

- $F_{\Phi_1} + F_{\Phi_2} = F_{\Phi_1 + \Phi_2} \in S_{\text{shape}}$,

- $aF_{\Phi_1} = F_{a\Phi_1} \in S_{\text{shape}}$.

This completes the proof.

$\square$

**Theorem B.8.** *Assume $S$ and $\mathcal{X}'$ are finite and $\Omega^*$ is proper, lower semicontinuous, and convex. Let $[r] := \{r + F_\Phi : \Phi \in \mathbb{R}^S\}$ denote the PBRS equivalence class of $r$. Assume that $\Omega^*$ is level-bounded on $[r]$, i.e., for every $c \in \mathbb{R}$ the set*

$$\{\, r' \in [r] : \Omega^*(r') \leq c \,\} \quad \text{is bounded.}$$

*Then the minimization $\min_{r' \in [r]} \Omega^*(r')$ exists, equivalently, $\mathrm{Proj}(r)$ is nonempty.*

*If additionally $\Omega^*$ is strictly convex, then the projected reward $\mathrm{Proj}(r)$ is unique.*

*Proof.* Since $\Omega^*$ is proper, lower semicontinuous, and convex, it is a closed convex function on the finite-dimensional space $\mathbb{R}^{\mathcal{X}'}$. The set $[r] = r + S_{\text{shape}}$ is a nonempty affine set (it contains $r$) and is closed and convex because $S_{\text{shape}}$ is a linear subspace as proved in Lemma B.7. By the assumed level-boundedness on $[r]$, each sublevel set

$$L_c := \{\, r' \in [r] : \Omega^*(r') \leq c \,\}$$

is bounded. Since $\Omega^*$ is lower semicontinuous and $[r]$ is closed, $L_c$ is also closed, hence compact in finite dimensions. Therefore, the infimum of $\Omega^*$ over $[r]$ is attained on $L_c$ for any $c$ above the infimum, and $\arg\min_{r' \in [r]} \Omega^*(r')$ is nonempty.

If $\Omega^*$ is strictly convex, then its restriction to the convex set $[r]$ is strictly convex. Hence, there can be at most one minimizer on $[r]$. Therefore $\mathrm{Proj}(r)$ is unique. $\square$

Beyond the conceptual motivation in the main paper, namely that the $\Omega^*$-aligned projection yields a consistent and conservative canonicalization, the projection can also improve inner optimization of the reward-robust RL problem from a max-min perspective. A growing line of work solves the inner minimization using dimension-free convex optimization methods such as mirror descent (Zhan et al., 2023; Bossens & Nitanda, 2025; Tomar et al., 2020). For these methods, iteration complexity typically depends on the size of the feasible set. In our setting, the projection selects, within each PBRS equivalence class, the representative that is smallest under the $\Omega^*$-geometry. As a consequence, projection cannot increase the $\Omega^*$-radius of the uncertainty set, which directly tightens the corresponding mirror-descent convergence bound. We first prove that the projection cannot increase the $\Omega^*$-radius of the uncertainty set.

**Proposition B.9.** *Assume $\Omega^* : \mathcal{R} \to \mathbb{R}$ is convex and bounded below. Define $\mathrm{Rad}_{\Omega^*}(\mathcal{U}) := \sup_{r \in \mathcal{U}} \Omega^*(r)$. Then for any uncertainty set $\mathcal{U} \subseteq \mathcal{R}$,*

$$\mathrm{Rad}_{\Omega^*}(\mathrm{Proj}(\mathcal{U})) \leq \mathrm{Rad}_{\Omega^*}(\mathcal{U}).$$

*Proof.* Fix any $r \in \mathcal{R}$. By definition,

$$\mathrm{Proj}(r) \in \arg\min_\Phi \Omega^*\big(r - (\gamma\Phi(s') - \Phi(s))\big).$$

Since $\Phi \equiv 0$ is feasible, we have $\Omega^*(\mathrm{Proj}(r)) \leq \Omega^*(r)$. Taking the supremum over $r \in \mathcal{U}$ yields

$$\sup_{\tilde{r} \in \mathrm{Proj}(\mathcal{U})} \Omega^*(\tilde{r}) = \sup_{r \in \mathcal{U}} \Omega^*(\mathrm{Proj}(r)) \leq \sup_{r \in \mathcal{U}} \Omega^*(r).$$

This completes the proof. $\square$

Since standard mirror-descent bounds scale with a geometry-dependent size term of the feasible set (e.g., a Bregman radius under the mirror map), reducing $\mathrm{Rad}_{\Omega^*}$ tightens the worst-case iteration complexity constants for the inner minimization. To make this dependence explicit, consider minimizing a convex function $f$ over a convex set $\mathcal{K}$ using mirror descent with

mirror map $\Psi$ that is $\rho$-strongly convex with respect to a norm $\| \cdot \|$. Assume the subgradients are bounded as $\|g_t\|_* \leq L$ for all iterations. A standard guarantee (e.g., Theorem 4.3 of (Bubeck et al., 2015)) states that with a suitable step size,

$$f(\bar{x}_t) - f(x^\star) \ \leq \ L\sqrt{\frac{2D}{\rho\, t}}, \qquad D := \sup_{x \in \mathcal{K}} D_\Psi(x, x_1),$$

where $D_\Psi$ is the Bregman radius induced by $\Psi$ and $x_1$ is the initialization. Equivalently, to reach accuracy $f(\bar{x}_t) - f(x^\star) \leq \delta$, it suffices that

$$t \ \geq \ \frac{2L^2}{\rho\, \delta^2}\, D.$$

In our setting, we take $\Psi = \Omega^*$ (we additionally assume $\Omega^*$ is $\rho$-strongly convex) and initialize at $x_1 = 0$.[1] Then the domain-size term becomes

$$D \ = \ \sup_{x \in \mathcal{K}} D_{\Omega^*}(x, 0) \ = \ \sup_{x \in \mathcal{K}} \Omega^*(x) \ =: \ \mathrm{Rad}_{\Omega^*}(\mathcal{K}).$$

Therefore, the mirror-descent iteration complexity scales linearly with $\mathrm{Rad}_{\Omega^*}(\mathcal{K})$:

$$t(\delta) \ \geq \ \frac{2L^2}{\rho\, \delta^2}\, \mathrm{Rad}_{\Omega^*}(\mathcal{K}).$$

As proved in Proposition B.9, for any feasible set $\mathcal{K} \subseteq \mathcal{R}$, projection cannot increase the $\Omega^*$-radius:

$$\mathrm{Rad}_{\Omega^*}(\mathrm{Proj}(\mathcal{K})) \ \leq \ \mathrm{Rad}_{\Omega^*}(\mathcal{K}).$$

Combining with the bound above, optimizing the inner problem over the projected set tightens the worst-case mirror-descent complexity guarantee, implying that fewer inner-loop iterations suffice to reach a target accuracy.

Since the $\Omega^*$-induced geometry can be abstract in full generality, we now illustrate the projection in a simple and concrete setting. Specifically, we consider the Hilbert space case and $\Omega^*$ is the squared $L^2$ norm. In this case, learning the projection reduces to fitting the shaping network (i.e., the potential $\Phi_\varphi$) by solving:

$$\varphi^* = \arg\min_\varphi \Big( r(s, a, s') - \big(\gamma\Phi_\varphi(s') - \Phi_\varphi(s)\big) \Big)^2.$$

To further understand this, we will continue to use the simple MDP defined in Figure 1 as an example. Here, every transition from $s_1$ to $s_2$ is deterministic, so the shaping term is the same constant $F$ for both actions. Then the projection becomes to fit a single constant $F$ to best match the two rewards $r_1 = r(s_1, a_1, s_2)$ and $r_2 = r(s_1, a_2, s_2)$:

$$\min_F \ (r_1 - F)^2 + (r_2 - F)^2.$$

This has a unique solution

$$F^* = \frac{r_1 + r_2}{2}.$$

And then the projected rewards:

$$r_1' = r_1 - F^* = \frac{r_1 - r_2}{2}, \qquad r_2' = r_2 - F^* = \frac{r_2 - r_1}{2},$$

which is exactly the reward example in Figure 1b (right).

## B.9. Analysis of Algorithm 1

In this section, we justify our key design choices and provide an overall analysis of Algorithm 1.

---

[1]This holds after an affine normalization: replace $\Omega^*(x)$ by $\widetilde{\Omega}^*(x) := \Omega^*(x) - \Omega^*(0) - \langle \nabla\Omega^*(0), x \rangle$, which does not change the induced Bregman divergence up to this choice of origin.

**Pre-computing vs. on-policy learning of the projection.** Theoretically, Theorem 4.1 is stated for an ideal canonicalized base reward $r_{\text{core}} = \text{Proj}(r)$. In principle, one may approximate this projection offline prior to policy optimization. For example, one can collect a large dataset, solve the projection problem to obtain $r_{\text{core}}$, and then run a standard RL optimizer using this fixed projected reward. This approach is viable when the discrepancy $\Omega^*$ is fixed (e.g., $\ell_p$ constrained uncertainty) and one is willing to pay the upfront data collection cost.

However, offline pre-computation can be undesirable for two reasons. First, it can be data-inefficient, since obtaining an accurate projection may require a large coverage dataset that is collected solely for canonicalization. Second, our framework allows general choices of $\Omega^*$. In some instances $\Omega^*$ is fixed, while in others it is defined through a trajectory-weighted objective (e.g., entropy/KL-induced forms) that depends on the current policy and therefore changes as learning proceeds. In this latter case, a single pre-computed projection may become stale as the policy evolves.

To obtain an algorithm that is both data-efficient and applicable to both policy-independent and policy-dependent $\Omega^*$, we adopt the on-policy strategy in Algorithm 1. The same rollouts are used to update the projection estimate and perform the policy update, so no additional data collection is required. Importantly, if $\Omega^*$ depends on the policy, then the target projection itself changes over training. Estimating the projection from rollouts under the current policy allows the learned canonicalization to adapt online as $\pi$ evolves.

**Convergence Analysis.** Algorithm 1 can be shown to converge asymptotically using standard two-timescale stochastic approximation arguments (Borkar & Borkar, 2008; Gadot et al., 2024). We place the projection update on the fast timescale and the policy update on the slow timescale. Intuitively, the projection parameters track their near-optimal values for the current policy, so the policy updates evolve as if they were computed using an almost-converged projection. Under standard conditions, this separation of timescales yields convergence to a stationary point of the target objective. Next, we outlined the detailed analysis. We begin by stating the assumptions used throughout this section.

**Assumption B.10.** The iterates $\{\theta_i\}$ generated by the algorithm remain bounded almost surely:

$$\sup_i \|\theta_i\| < \infty \qquad \text{a.s.}$$

This assumption is standard to ensure the iteration is bounded.

**Assumption B.11.** For each fixed $\theta \in \Theta$, the Markov chain induced by $(\mu_0, \pi_\theta, P)$ on states is ergodic with unique stationary distribution, and the trajectory used in Algorithm 1 are sampled from the stationary regime (or from sufficiently long rollouts).

This assumption typically holds when the environment is a continuous task where every state remains reachable, and the policy continues to explore. Sampling from the stationary regime ensures unbiased stochastic gradients.

**Assumption B.12.** Let $l(\varphi, \theta) := \Omega^*_{\pi_\theta}(r - F_{\Phi_\varphi})$ denote the projection loss in Algorithm 1, and let $\ell(\varphi, \theta; \mathcal{D}_{\pi_\theta})$ denote its empirical estimate computed from a rollout dataset $\mathcal{D}_{\pi_\theta}$. The population objective $L(\varphi, \theta) := \mathbb{E}[l(\varphi, \theta)|\theta]$ (the expectation of $l$ over the data-generating distribution induced by the policy $\pi_\theta$) is $\mu$-strongly convex with $L_\varphi$-Lipschitz gradient in $\varphi$, uniformly over $\theta \in \Theta$, and $L_\theta$-Lipschitz gradient in $\theta$, uniformly over $\varphi \in \Psi$, hence admits a unique minimizer:

$$\varphi^*(\theta) := \arg\min_{\varphi \in \Psi} L(\varphi, \theta).$$

Moreover, the stochastic gradient satisfies:

$$\mathbb{E}\left[ \left\| \nabla_\varphi \ell(\varphi, \theta; \mathcal{D}_{\pi_\theta}) - \nabla_\varphi L(\varphi, \theta) \right\|^2 \,\Big|\, \varphi, \theta \right] \leq \sigma^2.$$

This assumption holds when $\Phi_\varphi$ is tabular or linear in parameters (i.e., $\Phi_\varphi = \phi(s)^\top \varphi$ for some features $\phi(s)$) since in this case, minimizing $L(\varphi, \theta)$ actually solves a strongly convex problem, and the minimizer $\varphi^*$ is unique. For other function classes, for example, neural networks, this assumption does not necessarily hold, and in practice, we run SGD and treat the resulting $\Phi$ as an approximate projection and empirically observe stable behavior. The stochastic gradient assumption ensures the SGD update is stable.

**Assumption B.13.** The policy class $\pi_\theta$ is differentiable in $\theta$, and $\nabla_\theta \bar{J}(\theta, \varphi) := J(\pi_\theta, r_\varphi) - \Omega(\pi_\theta)$ is Lipschitz in $(\theta, \varphi)$.

This usually holds for smooth policy-gradient surrogates, and PPO is one practical instantiation.

**Assumption B.14.** Let $t$ index every update, including inner $\varphi$ update and outer $\theta$ update. Stepsizes $\{\alpha_t\}$ (for $\phi$) and $\{\beta_t\}$ (for $\theta$) satisfy

$$\sum_t \alpha_t = \infty, \qquad \sum_t \alpha_t^2 < \infty, \qquad \sum_t \beta_t = \infty, \qquad \sum_t \beta_t^2 < \infty, \qquad \frac{\beta_t}{\alpha_t} \to 0.$$

This assumption make sure that Algorithm 1 satisfies a two timescale argument where we update $\varphi$ (by SGD on $l(\varphi,\theta)$) fast while update $\theta$ (by policy gradient for $\bar{J}(\theta,\varphi)$) slow so that $\varphi$ stays near $\varphi^*(\pi_\theta)$ and if $\pi_\theta \to \pi_{\theta^*}$, then $\varphi \to \varphi^*(\pi_{\theta^*})$.

Next, before presenting the convergence proof, we first establish several supporting lemmas.

**Lemma B.15.**
$$\|a - b\|^2 \leq (1 + \eta)\|a\|^2 + \left(1 + \frac{1}{\eta}\right)\|b\|^2, \qquad \forall\, \eta > 0.$$

*Proof.* Notice that

$$\|a - b\|^2 = \|a\|^2 + \|b\|^2 - 2\langle a, b\rangle \leq \|a\|^2 + \|b\|^2 + 2|\langle a, b\rangle| \leq \|a\|^2 + \|b\|^2 + 2\|a\|\,\|b\|.$$

Consider the nonnegative square

$$\left(\sqrt{\eta}\, u - \frac{1}{\sqrt{\eta}}\, v\right)^2 \geq 0.$$

Expanding gives

$$\eta u^2 + \frac{1}{\eta} v^2 - 2uv \geq 0 \quad \Longrightarrow \quad 2uv \leq \eta u^2 + \frac{1}{\eta} v^2.$$

Let $u = \|a\|, v = \|b\|$:

$$2\|a\|\,\|b\| \leq \eta\|a\|^2 + \frac{1}{\eta}\|b\|^2.$$

Hence

$$\|a - b\|^2 \leq (1 + \eta)\|a\|^2 + \left(1 + \frac{1}{\eta}\right)\|b\|^2.$$

This completes the proof. $\square$

**Lemma B.16.** *Under Assumption B.12, there exists some constant $K$, such that*

$$\left\|\varphi^\star(\theta_1) - \varphi^\star(\theta_2)\right\| \leq K\left\|\theta_1 - \theta_2\right\|, \qquad \forall\, \theta_1, \theta_2.$$

*Proof.* Under Assumption B.12, there exists $\mu > 0$ such that for all $\theta$ and all $\varphi_1, \varphi_2$,

$$\left\langle \nabla_\varphi L(\varphi_1, \theta) - \nabla_\varphi L(\varphi_2, \theta),\, \varphi_1 - \varphi_2 \right\rangle \geq \mu\left\|\varphi_1 - \varphi_2\right\|^2.$$

And there exists $L_\theta > 0$ such that for all $\varphi$ and all $\theta_1, \theta_2$,

$$\left\|\nabla_\varphi L(\varphi, \theta_1) - \nabla_\varphi L(\varphi, \theta_2)\right\| \leq L_\theta\left\|\theta_1 - \theta_2\right\|.$$

Fix any $\theta_1, \theta_2$ and set

$$\varphi_1 := \varphi^\star(\theta_1), \qquad \varphi_2 := \varphi^\star(\theta_2).$$

Because $\varphi_1$ and $\varphi_2$ minimize $L(\cdot, \theta_1)$ and $L(\cdot, \theta_2)$, they satisfy the first-order optimality conditions:

$$\nabla_\varphi L(\varphi_1, \theta_1) = 0, \qquad \nabla_\varphi L(\varphi_2, \theta_2) = 0.$$

Consider

$$\begin{aligned}
0 &= \nabla_\varphi L(\varphi_1, \theta_1) - \nabla_\varphi L(\varphi_2, \theta_2) \\
&= \underbrace{\left(\nabla_\varphi L(\varphi_1, \theta_1) - \nabla_\varphi L(\varphi_2, \theta_1)\right)}_{(1)} + \underbrace{\left(\nabla_\varphi L(\varphi_2, \theta_1) - \nabla_\varphi L(\varphi_2, \theta_2)\right)}_{(2)}.
\end{aligned}$$

Take the inner product of both sides with $\varphi_1 - \varphi_2$:

$$\langle (1), \varphi_1 - \varphi_2 \rangle = -\langle (2), \varphi_1 - \varphi_2 \rangle.$$

Lower bound the left-hand side:

$$\langle (1), \varphi_1 - \varphi_2 \rangle = \langle \nabla_\varphi L(\varphi_1, \theta_1) - \nabla_\varphi L(\varphi_2, \theta_1), \varphi_1 - \varphi_2 \rangle \geq \mu \|\varphi_1 - \varphi_2\|^2.$$

Upper bound the right-hand side:

$$\begin{aligned}
\left| \langle (2), \varphi_1 - \varphi_2 \rangle \right| &\leq \|(2)\| \, \|\varphi_1 - \varphi_2\| \\
&= \left\| \nabla_\varphi L(\varphi_2, \theta_1) - \nabla_\varphi L(\varphi_2, \theta_2) \right\| \|\varphi_1 - \varphi_2\| \\
&\leq L_\theta \, \|\theta_1 - \theta_2\| \, \|\varphi_1 - \varphi_2\|.
\end{aligned}$$

Combining above gives

$$\mu \|\varphi_1 - \varphi_2\|^2 \leq L_\theta \, \|\theta_1 - \theta_2\| \, \|\varphi_1 - \varphi_2\|.$$

If $\varphi_1 = \varphi_2$, the claim is trivial. Otherwise, divide both sides by $\|\varphi_1 - \varphi_2\|$ to obtain

$$\|\varphi_1 - \varphi_2\| \leq \frac{L_\theta}{\mu} \|\theta_1 - \theta_2\| = K \|\theta_1 - \theta_2\|.$$

This completes the proof. $\square$

**Theorem B.17.** *Consider Algorithm 1 and suppose Assumptions B.10, B.11, B.12, B.13 and B.14 hold. Then, the iterates $(\theta_i, \varphi_i)$ generated by Algorithm 1 satisfy, almost surely:*

1. *The iterate for $\varphi$ tracks the instantaneous minimizer:*

$$\|\varphi_i - \varphi^*(\theta_i)\| \to 0, \qquad \varphi^*(\theta) := \arg\min_\varphi L(\varphi, \theta).$$

2. *Define the mean drift as $h(\theta, \varphi) := \mathbb{E}[\nabla_\theta \bar{J}(\theta, \varphi) | \theta, \varphi]$ The iterate $\{\theta_i\}$ is an asymptotic pseudo-trajectory of the ODE*

$$\dot{\theta} = h(\theta, \varphi^*(\theta)),$$

   *hence every limit point of $\{\theta_i\}$ lies in an internally chain transitive invariant set of this ODE.*

3. *If the mean $\theta$-update is an exact gradient field (i.e., the stochastic gradient estimator is unbiased so that the expected update $h(\theta, \varphi)$ equals $\nabla_\theta \bar{J}(\theta, \varphi)$), then every limit point $\theta_\infty$ satisfies*

$$\nabla_\theta \bar{J}(\theta_\infty, \varphi^*(\theta_\infty)) = 0.$$

*Proof.* 1. For fixed outer index $i$, define the gradient noise at inner step $k$:

$$M_{i,k+1} := \nabla_\varphi \ell(\varphi_{i,k}, \theta_i; D_{\pi_{\theta_i}}) - \nabla_\varphi L(\varphi_{i,k}, \theta_i).$$

By Assumption B.12, we have $\mathbb{E}\left[ \|M_{i,k+1}\|^2 \mid \varphi_{i,k}, \theta_i \right] \leq \sigma^2$. Then the inner update can be written as

$$\varphi_{i,k+1} = \varphi_{i,k} - \alpha_{i,k}\left( \nabla_\varphi L(\varphi_{i,k}, \theta_i) + M_{i,k+1} \right),$$

which is standard SGD on $L(\cdot, \theta_i)$. Let $e_{i,k} := \varphi_{i,k} - \varphi^*(\theta_i)$. Since $\varphi^*(\theta_i)$ minimizes $L(\cdot, \theta_i)$, we have $\nabla_\varphi L(\varphi^*(\theta_i), \theta_i) = 0$. Then we have,

$$\begin{aligned}
\|e_{i,k+1}\|^2 &= \left\| \varphi_{i,k+1} - \varphi^*(\theta_i) \right\|^2 \\
&= \left\| \varphi_{i,k} - \varphi^*(\theta_i) - \alpha_{i,k}\left( \nabla_\varphi L(\varphi_{i,k}, \theta_i) + M_{i,k+1} \right) \right\|^2 \\
&= \left\| e_{i,k} - \alpha_{i,k}\left( \nabla_\varphi L(\varphi_{i,k}, \theta_i) + M_{i,k+1} \right) \right\|^2
\end{aligned}$$

Take the conditional expectation given $\mathcal{F}_{i,k} := \sigma\big(\theta_i, \varphi_{i,0}, \ldots, \varphi_{i,k}, D_{\pi_{\theta_i}}\big)$. We obtain:

$$
\begin{aligned}
\mathbb{E}\big[\|e_{i,k+1}\|^2 \mid \mathcal{F}_{i,k}\big] &= \|e_{i,k}\|^2 - 2\alpha_{i,k}\big\langle e_{i,k}, \nabla_\varphi L(\varphi_{i,k}, \theta_i)\big\rangle - 2\alpha_{i,k}\big\langle e_{i,k}, \mathbb{E}\big[M_{i,k+1} \mid \mathcal{F}_{i,k}\big]\big\rangle \\
&\quad + \alpha_{i,k}^2\big\|\nabla_\varphi L(\varphi_{i,k}, \theta_i)\big\|^2 + \alpha_{i,k}^2\,\mathbb{E}\big[\|M_{i,k+1}\|^2 \mid \mathcal{F}_{i,k}\big]. \\
&\overset{(i)}{=} \|e_{i,k}\|^2 - 2\alpha_{i,k}\big\langle e_{i,k}, \nabla_\varphi L(\varphi_{i,k}, \theta_i)\big\rangle \\
&\quad + \alpha_{i,k}^2\big\|\nabla_\varphi L(\varphi_{i,k}, \theta_i)\big\|^2 + \alpha_{i,k}^2\,\mathbb{E}\big[\|M_{i,k+1}\|^2 \mid \mathcal{F}_{i,k}\big]. \\
&\overset{(ii)}{\leq} \big(1 - 2\mu\alpha_{i,k} + L_\varphi^2\alpha_{i,k}^2\big)\|e_{i,k}\|^2 + \sigma^2\alpha_{i,k}^2
\end{aligned}
$$

(i) uses $\mathbb{E}[M_{i,k+1} \mid \mathcal{F}_{i,k}] = 0$ (unbiased stochastic gradient under stationary sampling from Assumption B.11); (ii) use the $\mu$-strong convexity, $L_\varphi$-Lipschitz and bounded second moment of stochastic gradient from Assumption B.12

$$
\big\langle e_{i,k}, \nabla_\varphi L(\varphi_{i,k}, \theta_i)\big\rangle = \big\langle e_{i,k}, \nabla_\varphi L(\varphi_{i,k}, \theta_i) - \nabla_\varphi L(\varphi^*(\theta_i), \theta_i)\big\rangle \geq \mu\|e_{i,k}\|^2,
$$

$$
\big\|\nabla_\varphi L(\varphi_{i,k}, \theta_i)\big\| = \big\|\nabla_\varphi L(\varphi_{i,k}, \theta_i) - \nabla_\varphi L(\varphi^*(\theta_i), \theta_i)\big\| \leq L_\varphi\|e_{i,k}\|.
$$

Because $\sum_k \alpha_{i,k}^2 < \infty$ (Assumption B.14) and $\sum_k \alpha_{i,k} = \infty$, the Robbins-Siegmund supermartingale theorem (Robbins & Siegmund, 1971) implies

$$
\|e_{i,k}\|^2 \longrightarrow 0 \qquad \text{a.s. as } k \to \infty \quad \text{(for fixed } \theta_i\text{)}.
$$

Next, according to Assumption B.14, when we use $t$ index every update, then $\frac{\beta_t}{\alpha_t} \to 0$ means that $\theta$ moves so slowly that $\varphi$ has effectively infinite time to equilibrate between meaningful changes in $\theta$. Define $e_t := \varphi_t - \varphi^*(\theta_t)$, where now $\theta$ changes over time, so the optimum is moving. Now when we update $\varphi$ at time $t$, the next error is $e_{t+1} = \varphi_{t+1} - \varphi^*(\theta_{t+1}) = (\varphi_{t+1} - \varphi^*(\theta_t)) + (\varphi^*(\theta_t) - \varphi^*(\theta_{t+1}))$. Now use the inequality from Lemma B.15:

$$
\|a - b\|^2 \leq (1 + \eta)\|a\|^2 + \left(1 + \frac{1}{\eta}\right)\|b\|^2, \qquad \forall\,\eta > 0.
$$

Apply this with $a = \varphi_{t+1} - \varphi^*(\theta_t)$, $b = \varphi^*(\theta_t) - \varphi^*(\theta_{t+1})$):

$$
\|e_{t+1}\|^2 \leq (1 + \eta)\|\varphi_{t+1} - \varphi^*(\theta_t)\|^2 + \left(1 + \frac{1}{\eta}\right)\|\varphi^*(\theta_t) - \varphi^*(\theta_{t+1}))\|^2.
$$

Take conditional expectation given $\mathcal{F}_t$:

$$
\mathbb{E}\big[\|e_{t+1}\|^2 \mid \mathcal{F}_t\big] \leq (1 + \eta)\,\mathbb{E}\big[\|\varphi_{t+1} - \varphi^*(\theta_t)\|^2 \mid \mathcal{F}_t\big] + \left(1 + \frac{1}{\eta}\right)\|\varphi^*(\theta_t) - \varphi^*(\theta_{t+1}))\|^2.
$$

As for bounding $\mathbb{E}\big[\|\varphi_{t+1} - \varphi^*(\theta_t)\|^2 \mid \mathcal{F}_t\big]$, i.e., the error to the old minimizer $\varphi^*(\theta_t)$, based on the previous results regarding $e_{i,k+1}$, we have that there exist constant $c_1, c_2, c_3 > 0$, such that

$$
\mathbb{E}\big[\|\varphi_{t+1} - \varphi^*(\theta_t)\|^2 \mid \mathcal{F}_t\big] \leq \big(1 - c_1\alpha_t + c_2\alpha_t^2\big)\|e_t\|^2 + c_3\alpha_t^2
$$

Next, we bound $\|\varphi^*(\theta_t) - \varphi^*(\theta_{t+1}))\|^2$. Under Lemma B.16, $\varphi^*(\theta)$ is Lipschitz:

$$
\|\varphi^*(\theta') - \varphi^*(\theta)\| \leq K\|\theta' - \theta\| \quad \text{for some } K > 0.
$$

So

$$
\|\varphi^*(\theta_t) - \varphi^*(\theta_{t+1}))\| \leq K\|\theta_{t+1} - \theta_t\|.
$$

But $\theta$ changes with step size $\beta_t$, i.e.,

$$
\|\theta_{t+1} - \theta_t\| = O(\beta_t).
$$

Hence we have:

$$
\|\varphi^*(\theta_t) - \varphi^*(\theta_{t+1}))\|^2 \leq c_4\beta_t^2, \qquad \text{for some } c_4 > 0.
$$

Now apply Robbins-Siegmund again. Since $\sum_t \alpha_t^2 < \infty$ and $\sum_t \beta_t^2 < \infty$, the error terms are summable, and because $\sum_t \alpha_t = \infty$, we conclude that:

$$\|e_{t+1}\| \longrightarrow 0 \quad \text{a.s.}$$

Once we have $\|e_{t+1}\| \to 0$ for the global time $t$, it implies that at the subsequence of times corresponding to the end of each outer iteration (i.e., when we set $\varphi_{i+1} = \varphi_{i,K_i}$), we also have

$$\|\varphi_i - \varphi^*(\theta_i)\| \longrightarrow 0 \quad \text{a.s.}$$

2. The $\theta$-update can be written as

$$\theta_{i+1} = \theta_i + \beta_i\big(h(\theta_i, \varphi_{i+1}) + \xi_{i+1}\big),$$

where $\{\xi_{i+1}\}$ is a noise term:

$$\xi_{i+1} := \nabla_\theta \bar{J}(\theta_i, \varphi_{i+1}; D'_{\pi_{\theta_i}}) - h(\theta_i, \varphi_{i+1}).$$

The Lipschitz assumption on $\nabla_\theta \bar{J}(\theta, \varphi)$ (Assumption B.13) implies that $h$ is Lipschitz in $(\theta, \varphi)$ on the bounded region visited by the iterates since expectation preserves Lipschitzness. Boundedness of $\{\theta_i\}$ in Assumption B.10 and also $\sum_i \beta_i^2 < \infty$ (Assumption B.14) ensures that the accumulated noise is controlled in the ODE method. Now use $\|\varphi_{i+1} - \varphi^*(\theta_i)\| \to 0$ a.s. By Lipschitzness of $h$,

$$\big\|h(\theta_i, \varphi_{i+1}) - h(\theta_i, \varphi^*(\theta_i))\big\| \leq L_h \|\varphi_{i+1} - \varphi^*(\theta_i)\| \longrightarrow 0 \quad \text{a.s.}$$

Hence the $\theta$-recursion is an SA scheme with vanishing bias:

$$\theta_{i+1} = \theta_i + \beta_i\Big(h(\theta_i, \varphi^*(\theta_i)) + o(1) + \xi_{i+1}\Big).$$

Standard ODE-method conclusions (Kushner-Clark two-timescale SA (Borkar, 2008; Borkar & Meyn, 2000)) then imply that the continuous-time interpolation of $\{\theta_i\}$ is an asymptotic pseudo-trajectory of the limiting ODE

$$\dot{\theta} = h\big(\theta, \varphi^*(\theta)\big).$$

3. If we assume $h\big(\theta, \varphi^*(\theta)\big) = \nabla_\theta \bar{J}(\theta, \varphi)$, then the limiting ODE is

$$\dot{\theta} = \nabla_\theta \bar{J}\big(\theta, \varphi^*(\theta)\big).$$

Use $\bar{J}(\theta)$ short for $J\big(\theta, \varphi^*(\theta)\big)$, consider any solution trajectory $\theta(t)$ of this ODE. By the chain rule,

$$\frac{d}{dt}\, \bar{J}(\theta(t)) = \big\langle \nabla \bar{J}(\theta(t)), \dot{\theta}(t) \big\rangle.$$

Substituting $\dot{\theta}(t) = \nabla \bar{J}(\theta(t))$ yields

$$\frac{d}{dt}\, \bar{J}(\theta(t)) = \big\langle \nabla \bar{J}(\theta(t)), \nabla \bar{J}(\theta(t)) \big\rangle = \big\|\nabla \bar{J}(\theta(t))\big\|^2 \geq 0.$$

Therefore $\bar{J}(\theta(t))$ is non-decreasing along trajectories, and the only points where $\bar{J}$ stops increasing are those for which $\nabla \bar{J}(\theta) = 0$. Hence every limit point of the ODE must lie in the set of stationary points

$$\{\theta : \nabla \bar{J}(\theta) = 0\}.$$

Since the stochastic iterates $\{\theta_i\}$ form an asymptotic pseudo-trajectory of the ODE, their limit points are contained in the same stationary set, i.e.,

$$\nabla \bar{J}(\theta_\infty) = 0.$$

Equivalently,

$$\nabla_\theta \bar{J}\big(\theta_\infty, \varphi^*(\theta_\infty)\big) = 0,$$

This completes the proof. $\qquad\square$

# C. Additional Implementation Details

All experiments were conducted on a single NVIDIA RTX 4090 GPU (24GB memory) and a 13th Gen Intel Core i9-13900KF CPU (32 threads). We implemented all methods in Python 3.9 using PyTorch 2.6.0 (Paszke, 2019), and SB3 (Raffin et al., 2021).

## C.1. Implementation Details for MiniGrid

We use a simple deterministic $N \times N$ grid-world task as in previous work (Ashlag et al., 2025; Chevalier-Boisvert et al., 2018). As shown in Figure 2, the agent starts in the top-left cell and aims to reach a fixed goal in the bottom-right cell. The action space consists of four deterministic moves: up, down, left, and right. Attempting to move outside the grid leaves the agent in the same cell. The goal state is terminal and absorbing. Rewards are defined on deterministic transitions. Each non-terminal step incurs a constant step cost of $-0.01$. In addition, the agent receives a 1 reward upon entering the goal cell. Once at the goal, all subsequent rewards are zero. We use a discount factor $\gamma = 0.99$ and an initial state distribution concentrated on the start state.

For the robust training, we consider an $l_2$-bounded perturbation set over state-action rewards. Concretely, letting $r$ denote the original reward and $\delta$ as additive perturbation, we define:

$$\mathcal{U} = \{r + \delta : \|\delta\|_2 \leq \varepsilon\} \subset \mathbb{R}^{\mathcal{X}'}.$$

Then the reward-robust objective is the standard max-min problem:

$$\max_{\pi} \min_{\tilde{r} \in \mathcal{U}} J(\pi, \tilde{r})$$

To optimize this objective, we use a standard game-style procedure. At each outer iteration $i$, given the current policy $\pi_i$, we approximately solve the inner minimization by projected gradient steps on the reward perturbation $\delta$, descending the return $J(\pi_i, r + \delta)$ while projecting back onto the feasible set $\{\delta : \|\delta\|_2 \leq \varepsilon\}$. This yields an approximate worst-case reward $\tilde{r}_i \in \mathcal{U}$. We then perform a policy-optimization step via solving the Bellman equations to increase $J(\pi, \tilde{r}_i)$, producing the next policy $\pi_{i+1}$. Repeating these two steps implements a practical saddle-point optimization of the robust objective.

As for implementing our shaping-aware method, recall that the shaping-aware uncertainty set is defined as the image of the original uncertainty set under the PBRS canonicalization map:

$$\mathcal{U}_{\text{core}} := \text{Proj}(\mathcal{U}) = \left\{\text{Proj}(r + \delta) : \|\delta\|_2 \leq \varepsilon\right\}.$$

In our grid-world, transitions are deterministic, so the rewards can be represented as a state-action vector $r \in \mathbb{R}^{\mathcal{X}}$. We use the $l_2$ geometry in our uncertainty set, consequently, $\text{Proj}$ is an orthogonal projection that removes PBRS components.

Let $\Phi \in \mathbb{R}^{\mathcal{S}}$ be a potential function and define the shaping operator $B \in \mathbb{R}^{\mathcal{S}\mathcal{A} \times \mathcal{S}}$ by

$$(B\Phi)(s, a, s') := \gamma\Phi(s') - \Phi(s),$$

where $s'$ is the deterministic next state after taking action $a$ in state $s$. The shaping subspace is $\mathcal{S}_{\text{shape}} = \text{range}(B)$. To make the decomposition unique, we fix a gauge (e.g., $\Phi(s_0) = 0$ ). The projection of a reward $r$ is then

$$\text{Proj}(r) = r - B\Phi^*(r), \qquad \Phi^*(r) \in \arg\min_{\Phi:\,\Phi(s_0)=0} \left\|r - B\Phi\right\|_2^2.$$

Equivalently, $\text{Proj}(r)$ is linear and can be written as $\text{Proj}(r) = Pr$, where $P \in \mathbb{R}^{\mathcal{S}\mathcal{A} \times \mathcal{S}\mathcal{A}}$ is the orthogonal projector onto the complement of $\text{range}(B)$. We denote the nominal core reward by

$$r_{\text{core}} := \text{Proj}(r) = Pr.$$

A direct characterization of $\mathcal{U}_{\text{core}}$ is therefore:

$$\mathcal{U}_{\text{core}} = \left\{r_{\text{core}} + P\delta : \|\delta\|_2 \leq \varepsilon\right\},$$

Thus, the original $\ell_2$-ball uncertainty set becomes a linear image of that set under $P$: a convex polytope in the core reward space. Importantly, the projection collapses all PBRS-equivalent perturbations, i.e., all perturbations differing only by an element of $\mathrm{range}(B)$, to the same core reward.

In practice, rather than explicitly parameterizing $\delta$ and repeatedly applying $P$, we optimize the inner problem directly in a core coordinate system. Concretely, we precompute a matrix $Q \in \mathbb{R}^{SA \times d}$ whose columns form an orthonormal basis for the projected subspace (so that $P = QQ^\top$), and represent core rewards as $r_{\mathrm{core}} = Qz$ for $z \in \mathbb{R}^d$. The inner minimization is then carried out over $\tilde{z}$ with an $\ell_2$ budget:

$$\min_{\tilde{z}:\|\tilde{z}-z\|_2 \leq \varepsilon} J(\pi, Q\tilde{z}), \qquad z := Q^\top r,$$

using projected gradient descent in $\tilde{z}$-space. The outer maximization updates the policy parameters using the resulting worst-case core reward $Qz$. This yields the shaping-aware max-min solver, where every adversary update is confined to the behaviorally meaningful directions by construction.

For the numerical simulation, we use a softmax policy and set the uncertainty budget $\varepsilon = 5.0$. Both methods are trained for 300 outer policy iterations, using a policy learning rate of 0.5. For the inner adversary problem, we run 20 update steps per outer iteration with an adversary learning rate of 0.5. To quantify how much the inner adversary optimization spends updating PBRS-invariant directions, we measure, for each inner step, the fraction of the update that lies in the shaping subspace $\mathrm{range}(B)$. Define the complementary projector onto shaping directions by

$$P_{\mathrm{shape}} := I - P.$$

At outer iteration $i$, let $r^{(i,j)}$ be the reward after the $j$-th inner adversary step ($j = 0, \ldots, J$), and define the per-step update

$$\Delta r^{(i,j)} := r^{(i,j+1)} - r^{(i,j)}.$$

We then compute the PBRS-invariant update fraction for that step as

$$\alpha^{(i,j)} := \frac{\left\| P_{\mathrm{shape}} \Delta r^{(i,j)} \right\|_2}{\left\| \Delta r^{(i,j)} \right\|_2 + \epsilon},$$

where $\epsilon > 0$ is a small constant for numerical stability. Finally, we report the average fraction over the $J$ inner updates at iteration $i$:

$$\mathrm{Frac}(i) := \frac{1}{J} \sum_{j=0}^{J-1} \alpha^{(i,j)}.$$

By construction, values $\mathrm{Frac}(i) \approx 1$ indicate that most adversary updates move along PBRS-equivalent directions, whereas values near 0 indicate that the updates primarily change the core reward.

**C.2. Implementation Details for MuJoCo**

For the MuJoCo experiments, we consider two reward-robust formulations introduced in Section 2. In both cases, we report results under the corresponding reward-robust objective

$$\max_\pi \min_{\tilde{r} \in \mathcal{U}} J(\pi, \tilde{r}),$$

and implement training via the equivalent regularized RL objective. We optimize this objective using PPO (Schulman et al., 2017), where the regularization term is included as an intrinsic reward. Notice that in the MuJoCo environments considered in our experiments, the per-step reward depends on the current state and action. We therefore write reward as $r(s, a)$ and omit the next state argument for simplicity.

**Entropy-based robust objective.** We first consider the entropy-induced uncertainty set

$$\mathcal{U} = \left\{ \tilde{r} : \mathbb{E}\left[ \sum_{t=0}^\infty \gamma^t \log \int_\mathcal{A} \exp\big(r(s_t, a) - \tilde{r}(s_t, a)\big)\, da \,\Big|\, \mu_0, \pi, P \right] \leq \epsilon \right\},$$

which is well known to be equivalent to optimizing an entropy-regularized RL objective:

$$\max_{\pi} \ J(\pi, r) \ - \ \mathbb{E}\left[\sum_{t=0}^{\infty} \gamma^t \int_{\mathcal{A}} \pi(a \mid s_t) \log \pi(a \mid s_t) \, da \,\middle|\, \mu_0, \pi, P\right],$$

following prior work (Haarnoja et al., 2018; Eysenbach & Levine, 2021).

For our shaping-aware variant, we add a projection step that learns the shaping potential function $\Phi_\varphi$ and replaces the reward with its PBRS-canonicalized version. For the entropy case, the projection loss is

$$\varphi^* \in \arg\min_{\varphi} l(\varphi) = \arg\min_{\varphi} \ \mathbb{E}_{\mathcal{D}}\left[\sum_{t=0}^{\infty} \gamma^t \ \log \int_{\mathcal{A}} \exp\Big(r(s_t, a) - \big(\gamma\Phi_\varphi(s_{t+1}) - \Phi_\varphi(s_t)\big)\Big) da\right].$$

We estimate the objective using a batch dataset $\mathcal{D}$ of $N$ on-policy trajectories. The inner $\log \int_{\mathcal{A}} \exp(\cdot) \, da$ term is an action-space log-partition function, which we approximate by Monte Carlo with a proposal density $q(\cdot \mid s)$. In our implementation we take $q(\cdot \mid s) = \pi(\cdot \mid s)$ (the current policy) and draw $K$ i.i.d. actions $a^{(1)}, \ldots, a^{(K)} \sim \pi(\cdot \mid s_t)$, yielding the importance-sampled estimator

$$\log \int_{\mathcal{A}} \exp(u_t(a)) \, da \ \approx \ \log\left(\frac{1}{K} \sum_{k=1}^{K} \frac{\exp(u_t(a^{(k)}))}{\pi(a^{(k)} \mid s_t)}\right),$$

where $u_t(a) = r(s_t, a) - (\gamma\Phi_\varphi(s_{t+1}) - \Phi_\varphi(s_t))$. Substituting this estimator into the projection objective yields the empirical minibatch loss

$$\widehat{l}(\varphi; \mathcal{D}) := \frac{1}{N} \sum_{i=1}^{N} \sum_{t=0}^{T_i-1} \gamma^t \ \log\left(\frac{1}{K} \sum_{k=1}^{K} \frac{\exp\big(u_t^{(i)}(a^{(i,k)})\big)}{\pi(a^{(i,k)} \mid s_t^{(i)})}\right), \qquad \varphi^* \in \arg\min_{\varphi} \ \widehat{l}(\varphi; \mathcal{D}).$$

**KL-based robust objective.** Next, we consider the KL-induced uncertainty set

$$\mathcal{U} = \left\{\tilde{r} : \ \mathbb{E}\left[\sum_{t=0}^{\infty} \tau\gamma^t \log \int_{\mathcal{A}} \pi_{\text{ref}}(a \mid s_t) \exp\left(\frac{r(s_t, a) - \tilde{r}(s_t, a)}{\tau}\right) \, da \,\middle|\, \mu_0, \pi, P\right] \le \varepsilon\right\},$$

where $\tau > 0$ is the regularization weight and $\pi_{\text{ref}}$ is a reference policy. This robust formulation corresponds to a KL-regularized RL objective (Table 3 in (Derman et al., 2023)):

$$\max_{\pi} \ J(\pi, r) \ - \ \tau \mathbb{E}\left[\sum_{t=0}^{\infty} \gamma^t \text{KL}\big(\pi(\cdot \mid s_t) \,\big\|\, \pi_{\text{ref}}(\cdot \mid s_t)\big) \,\middle|\, \mu_0, \pi, P\right].$$

For our shaping-aware variant, the projection loss becomes

$$\varphi^* \in \arg\min_{\varphi} l(\varphi) = \arg\min_{\varphi} \mathbb{E}_{\mathcal{D}}\left[\sum_{t=0}^{\infty} \gamma^t \log \int_{\mathcal{A}} \pi_{\text{ref}}(a \mid s_t) \exp\left(\frac{r(s_t, a) - \big(\gamma\Phi_\varphi(s_{t+1}) - \Phi_\varphi(s_t)\big)}{\tau}\right) da\right].$$

Similarly, we estimate the expectation using a batch dataset $\mathcal{D}$ of $N$ on-policy trajectories. Define

$$u_t(a) := \frac{r(s_t, a) - \big(\gamma\Phi_\varphi(s_{t+1}) - \Phi_\varphi(s_t)\big)}{\tau}.$$

We approximate the inner log term by Monte Carlo using a proposal density $q(\cdot \mid s)$. In our implementation, we take $q(\cdot \mid s) = \pi_{\text{ref}}(\cdot \mid s)$ to be the reference policy. Drawing $K$ i.i.d. actions $a^{(1)}, \ldots, a^{(K)} \sim \pi_{\text{ref}}(\cdot \mid s_t)$ yields the importance-sampled estimator

$$\log \int_{\mathcal{A}} \pi_{\text{ref}}(a \mid s_t) \exp\big(u_t(a)\big) \, da \ \approx \ \log\left(\frac{1}{K} \sum_{k=1}^{K} \exp\big(u_t(a^{(k)})\big)\right).$$

Substituting this estimator yields the empirical minibatch loss

$$\widehat{l}(\varphi; \mathcal{D}) := \frac{1}{N} \sum_{i=1}^{N} \sum_{t=0}^{T_i-1} \gamma^t \, \log\left( \frac{1}{K} \sum_{k=1}^{K} \exp\left( u_t^{(i)}(a^{(i,k)}) \right) \right), \qquad \varphi^* \in \arg\min_{\varphi} \, \widehat{l}(\varphi; \mathcal{D}).$$

Finally, to satisfy the technical conditions required by our convergence analysis (Assumption B.12), we add a small ridge term to the shaping loss,

$$\ell_{\lambda,\beta}(\varphi) := \ell(\varphi) + \frac{\lambda}{2} \|\varphi\|^2 + \beta \, \mathbb{E}_{s_0 \sim \mathcal{D}}\left[ \Phi_\varphi(s_0)^2 \right], \qquad \lambda > 0, \; \beta > 0,$$

where the $\beta$-term also encourages the gauge condition corresponding to $C(\Phi) = 0$ in our PBRS normalization. We parameterize $\Phi_\varphi$ by a two-layer MLP with 256 hidden units per layer and ReLU activations, and use the same architecture across all environments. For the reference policy $\pi_{\text{ref}}$, we use a uniform policy to ensure the baseline is not informative so that all methods have equal room for improvement. In our implementation, we fix the step size and the number of iterations for projection update. Full hyperparameters for the projection step and PPO are reported in Table 4 and Table 3.

*Table 3.* Hyperparameters used for projection across different environments.

| Hyperparameter | Hopper | Walker | HalfCheetah | Humanoid | Reacher |
|---|---|---|---|---|---|
| Iteration number | 10 | 5 | 5 | 5 | 5 |
| Batch size | 256 | 256 | 256 | 256 | 256 |
| Sampled actions $K$ | 64 | 64 | 64 | 64 | 64 |
| Optimizer | Adam | Adam | Adam | Adam | Adam |
| Step size | $1 \times 10^{-3}$ | $1 \times 10^{-4}$ | $3 \times 10^{-4}$ | $3 \times 10^{-4}$ | $3 \times 10^{-4}$ |
| Ridge term weight $\lambda$ | $1 \times 10^{-4}$ | $1 \times 10^{-4}$ | $1 \times 10^{-4}$ | $1 \times 10^{-4}$ | $1 \times 10^{-4}$ |
| Gauge term weight $\beta$ | $1 \times 10^{-3}$ | $1 \times 10^{-3}$ | $1 \times 10^{-3}$ | $1 \times 10^{-3}$ | $1 \times 10^{-3}$ |
| KL coefficient $\tau$ | 0.2 | 0.1 | 0.2 | 0.1 | 0.1 |

*Table 4.* Hyperparameters used for PPO training for all methods across different environments.

| Hyperparameter | Hopper | Walker | HalfCheetah | Humanoid | Reacher |
|---|---|---|---|---|---|
| Total timesteps | 500000 | 1000000 | 1000000 | 1000000 | 500000 |
| Batch size | 64 | 64 | 64 | 64 | 64 |
| Number steps | 2048 | 2048 | 2048 | 2048 | 2048 |
| Number envs | 1 | 1 | 1 | 1 | 1 |
| Update epochs | 10 | 10 | 10 | 10 | 10 |
| Optimizer | Adam | Adam | Adam | Adam | Adam |
| Learning rate | $3 \times 10^{-4}$ | $3 \times 10^{-4}$ | $3 \times 10^{-4}$ | $3 \times 10^{-4}$ | $3 \times 10^{-4}$ |
| Discount factor ($\gamma$) | 0.99 | 0.99 | 0.99 | 0.99 | 0.99 |
| Entropy coefficient | 0.01 | 0.01 | 0.001 | 0.001 | 0.01 |
| KL coefficient | 0.2 | 0.1 | 0.2 | 0.1 | 0.1 |

### C.3. Implementation Details for Reward Perturbation

For the locomotion tasks (HalfCheetah, Hopper, Walker2d, and Humanoid), during training, we construct the adversarial reward $r_{\text{adv}}$ using (Husain et al., 2021):

$$r_{\text{adv}} = \begin{cases} r(s, a) + \delta, & \text{if } r(s, a) \leq 0, \\ r(s, a), & \text{otherwise.} \end{cases}$$

where $\delta$ is drawn from a Gaussian distribution whose mean (the reward-shift radius) varies across conditions, while the variance is fixed to $0.1$. We found performance to be largely insensitive to this variance choice, so we keep it fixed and vary only the mean.

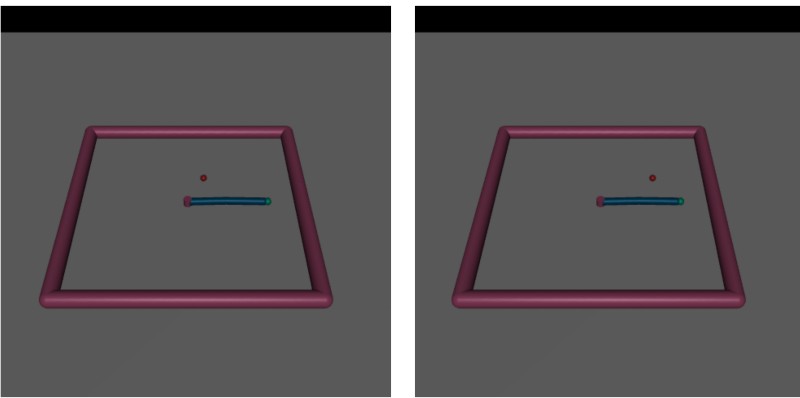

*Figure 5.* Example goal shift in Reacher. Left: the original environment. Right: the perturbed environment with a shifted goal.

For Reacher, we relocate the goal to a random position at varying distances from its nominal location during evaluation (Ashlag et al., 2025). Figure 5 illustrates the original Reacher environment and an example perturbed setting with a shifted goal.

### C.4. Implementation Details for Ablation Study

To isolate the contribution of our $\Omega^*$-aligned canonicalization, we compare against several alternative projection choices. First, we include a random $\Phi$ variant, where the potential network $\Phi_\varphi$ is kept at its random initialization (i.e., $\varphi$ is not trained). Second, we include an $L^2$ projection baseline that ignores the geometry induced by the uncertainty set and instead defines $\Omega^*$ as a squared $\ell_2$ norm. Concretely, for both the entropy and KL settings, this corresponds to fitting $\Phi_\varphi$ by:

$$\varphi^* \in \arg\min_\varphi \ l(\varphi) \ := \ \mathbb{E}_{\mathcal{D}}\left[\sum_{t=0}^{\infty} \gamma^t \left(r(s_t, a_t, s_{t+1}) - \left(\gamma\Phi_\varphi(s_{t+1}) - \Phi_\varphi(s_t)\right)\right)^2\right].$$

We estimate the objective using a batch dataset $\mathcal{D}$ of $N$ on-policy trajectories:

$$\widehat{l}(\varphi; \mathcal{D}) = \frac{1}{N}\sum_{i=1}^{N}\sum_{t=0}^{T_i-1} \gamma^t \left(r^{(i)}(s_t, a_t, s_{t+1}) - \left(\gamma\Phi_\varphi(s_{t+1}^{(i)}) - \Phi_\varphi(s_t^{(i)})\right)\right)^2,$$

## D. Additional Experiment Results

### D.1. Analysis on Non-convex Uncertainty Set

As discussed in the main text, when the adversary searches for a worst-case reward directly in the original reward space, it may spend many iterations moving along PBRS-invariant directions that do not change policy ranking. This issue is particularly acute for non-convex uncertainty sets (e.g., unions of scenarios) or implicitly specified feasible sets, where there may be no efficient dedicated solver for the optimization, and one must resort to black-box procedures (random search, Bayesian optimization, etc.) (Choromanski et al., 2020; Hüttenrauch & Neumann, 2024; Kirschner et al., 2020; Inatsu et al., 2022; Inatsu, 2025). Such methods typically suffer from sample complexity that scales badly with the dimension. Our shaping-aware projection reduces the intrinsic search dimension by removing PBRS-invariant directions, which can substantially improve the efficiency of black-box adversary search.

To illustrate, consider the same toy setting as in the main text in Figure 1 and define a non-convex uncertainty set as the union of $K$ thin diagonal bands:

$$\mathcal{U} \ = \ \bigcup_{k=1}^{K}\left\{(r_1, r_2) \in [0,1]^2 \ : \ \left|(r_1 - r_2) - \delta_k\right| \le \eta\right\}.$$

When $\eta$ is small, $\mathcal{U}$ consists of $K$ narrow strips aligned with the PBRS-invariant $(1, 1)$ direction (Figure 6). The union of several scenarios makes the feasible set highly non-convex. Under the projection that fixes one coordinate (e.g., $r_2^{\text{core}} = 0$),

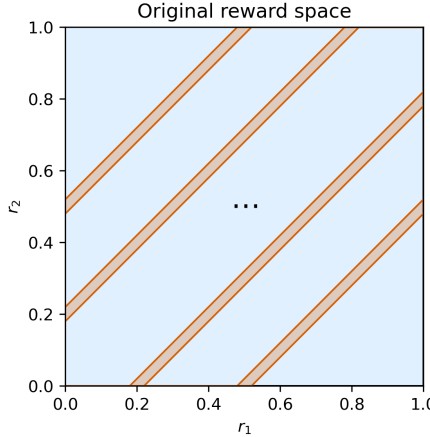

*Figure 6.* Illustration of a nonconvex uncertainty set.

the set collapses to a one-dimensional union of intervals:

$$\mathcal{U}_{\text{core}} \;=\; \bigcup_{k=1}^{K} [\delta_k - \eta, \; \delta_k + \eta] \;\subset\; [-1, 1].$$

Thus, projection removes the long PBRS-invariant direction, retaining only the behaviorally relevant coordinate.

Now suppose the adversary can only query feasible rewards. In the original two-dimensional set $\mathcal{U}$, a generic black-box exploration strategy will nevertheless allocate evaluations across both directions in the ambient space. Since rewards are behaviorally identical along the invariant direction, many of these evaluations are redundant. In contrast, searching over $\mathcal{U}_{\text{core}}$ allocates all evaluations along the identified coordinate, so each query is informative.

A simple scaling argument makes this difference explicit. Consider a single strip of width $2\eta$. In $\mathcal{U}_{\text{core}}$, choosing one reward that is within $\varepsilon$ of the true worst-case reward along the interval requires on the order of

$$N_{\mathcal{U}_{\text{core}}} \;\approx\; \frac{2\eta}{\varepsilon}$$

feasible evaluations. In the original reward space, the corresponding strip has area approximately

$$A \;\approx\; (2\eta)\, L, \qquad L \approx 1 - |\delta_k|$$

where $L$ is the length of the diagonal segment inside $[0, 1]^2$. For a two-dimensional space with $N$ points, the typical spacing scales as $h \approx \sqrt{A/N}$. To obtain comparable $\varepsilon$-level resolution, one needs

$$N_{\mathcal{U}} \;\approx\; \frac{A}{\varepsilon^2} \;\approx\; \frac{2\eta L}{\varepsilon^2}.$$

Therefore, black-box search scales as $\mathcal{O}(1/\varepsilon)$ over $\mathcal{U}_{\text{core}}$ but $\mathcal{O}(1/\varepsilon^2)$ over $\mathcal{U}$, with the extra factor arising from the algorithm implicitly spending samples to cover the PBRS-invariant direction.

For a numerical illustration, let $\eta = 0.01$, $\delta_k = 0.2$ (so $L \approx 0.8$), and $\varepsilon = 10^{-3}$. Then

$$N_{\mathcal{U}_{\text{core}}} \approx \frac{0.02}{0.001} = 20, \qquad N_{\mathcal{U}} \approx \frac{2\eta L}{\varepsilon^2} = \frac{0.02 \times 0.8}{10^{-6}} = 16{,}000.$$

In the original space, the overwhelming majority of evaluations are effectively spent spreading points along a behaviorally irrelevant direction. After projection, essentially every evaluation contributes to identifying a more adverse reward.

**D.2. Learning Curves on MuJoCo Tasks.**

Figure 7 reports the learning curves for both reward-robust objectives. Consistent with our main findings, incorporating shaping-aware projection improves final performance and typically reaches a given return with fewer environment interactions, indicating faster convergence than the standard reward-robust baseline.

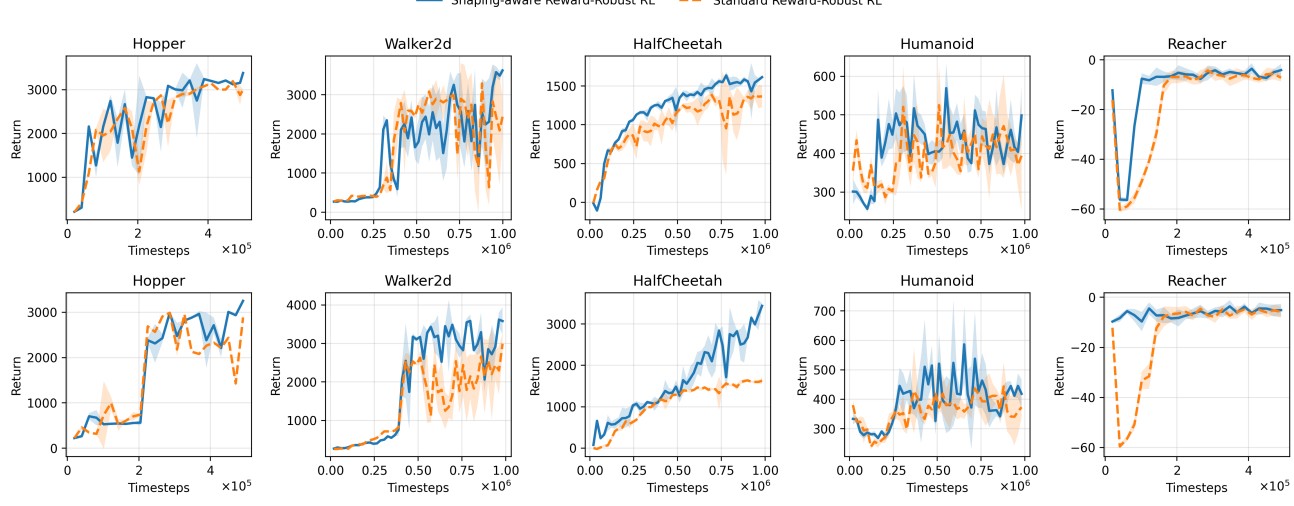

*Figure 7.* Mean and standard deviation return of the evaluation policy over 10 rollouts and 5 seeds, reported every 20k environment steps. The return is in terms of the environment's original reward. Top row: Entropy-based reward-robust problem, bottom row: KL-based reward-robust problem.

## D.3. Reward Robustness Results

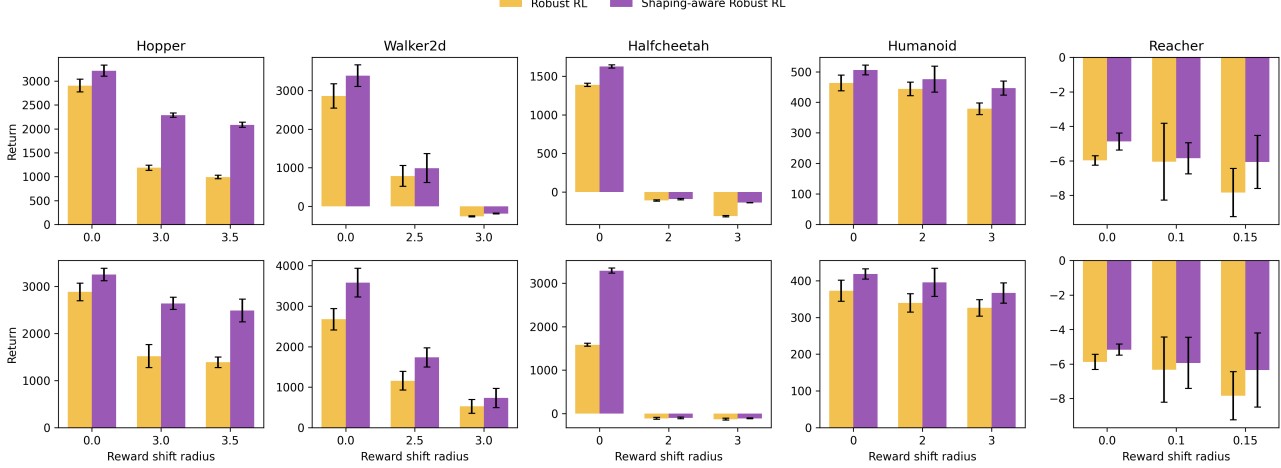

*Figure 8.* Mean ± 95% confidence interval return of the evaluation policy under reward perturbation over 100 episodes and 5 seeds. Top row: Entropy-based reward-robust problem, bottom row: KL-based reward-robust problem.

Figure 8 shows the robustness under reward perturbations for both reward-robust problems. Overall, the shaping-aware method exhibits smaller performance degradation than the robust RL baseline as the reward shift radius increases, indicating improved stability to reward misspecification.

## D.4. Analysis on Training Time

Table 5 reports the approximate training time for the standard reward-robust baseline and our shaping-aware variant. Across all five MuJoCo tasks, the shaping-aware method incurs a small additional cost (typically around 10–20 minutes), which is attributable to the extra projection step that fits the potential function. Importantly, this overhead is modest relative to the overall training time (on the order of around 10–25% depending on the environment) and remains practical in the same computational setting. Given the consistent gains in final return and faster convergence observed in Section 5, the added computation is a affordable trade-off for improved robustness and performance.

*Table 5.* Approximate training time for each algorithm across different continuous-control environments.

| Algorithm | Hopper | Walker | HalfCheetah | Humanoid | Reacher |
|---|---|---|---|---|---|
| Standard Reward-Robust RL | ≈50m | ≈1h50m | ≈1h30m | ≈1h40m | ≈ 50m |
| Shaping-aware Reward-Robust RL | ≈1h | ≈2h10m | ≈1h50m | ≈2h | ≈ 1h |

*Table 6.* Ablation on canonicalization choices under reward perturbations. For Hopper, Walker2d, HalfCheetah, and Humanoid, we use a reward-shift radius of 3. For Reacher, we use a goal-shift radius of 0.1. Results are reported as mean $\pm$ 95% confidence interval over 5 random seeds.

| Method | Hopper | Walker2d | HalfCheetah | Humanoid | Reacher |
|---|---|---|---|---|---|
| **Entropy-based robust objective** | | | | | |
| Reward-Robust (baseline) | 1186.66$\pm$52.27 | -259.59$\pm$16.07 | -313.50$\pm$9.73 | 378.97$\pm$18.90 | -6.05$\pm$2.22 |
| Shaping-aware (ours, $\Omega^*$-projection) | 2287.66$\pm$44.64 | -188.14$\pm$9.26 | -138.06$\pm$1.57 | 446.41$\pm$23.16 | -5.85$\pm$0.90 |
| Shaping-aware (random $\Phi$ canonicalization) | 1252.33$\pm$43.81 | -226.17$\pm$36.72 | -361.70$\pm$9.97 | 338.29$\pm$22.26 | -6.35$\pm$2.14 |
| Shaping-aware ($L^2$ projection) | 1491.77$\pm$37.04 | -199.04$\pm$9.37 | -144.56$\pm$6.25 | 474.09$\pm$40.15 | -5.94$\pm$1.48 |
| **KL-based robust objective** | | | | | |
| Reward-Robust (baseline) | 1519.33$\pm$241.86 | 525.17$\pm$171.17 | -128.73$\pm$21.13 | 325.81$\pm$22.22 | -6.33$\pm$1.89 |
| Shaping-aware (ours, $\Omega^*$-projection) | 2638.49$\pm$129.72 | 733.78$\pm$238.30 | -110.16$\pm$10.06 | 366.31$\pm$27.34 | -5.94$\pm$1.48 |
| Shaping-aware (random $\Phi$ canonicalization) | 1230.95$\pm$124.39 | 446.50$\pm$170.87 | -125.51$\pm$21.98 | 311.47$\pm$15.03 | -6.98$\pm$1.92 |
| Shaping-aware ($L^2$ projection) | 2016.28$\pm$113.72 | 693.48$\pm$109.97 | -111.75$\pm$13.99 | 335.10$\pm$14.12 | -6.17$\pm$1.41 |

## D.5. Additional Ablation Study

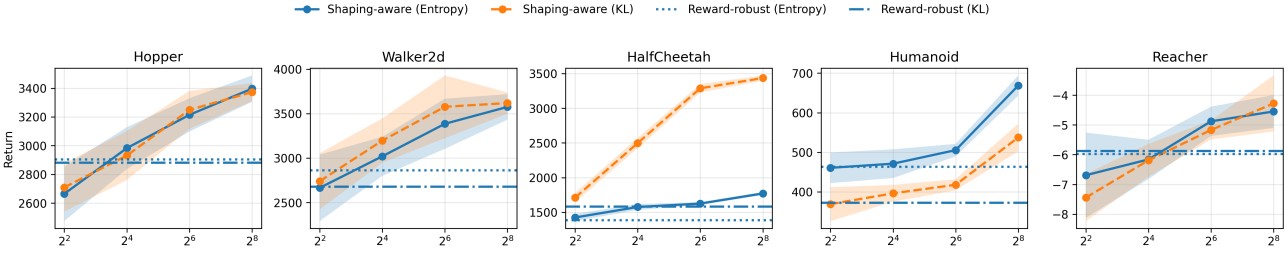

*Figure 9.* Ablations on sampled actions for projection learning. We evaluate each policy for 100 episodes and show 95% confidence interval across 5 seeds.

**Sensitivity to data coverage in the projection estimate.** As detailed in Appendix C, the $\Omega^*$ term used in our MuJoCo experiments involves a log-sum-exp over actions. Since the action space is continuous, we approximate this quantity via Monte Carlo by sampling $K$ actions (from the policy or the reference policy, depending on the objective). This approximation quality directly affects the projection step. When $K$ is small, the estimated projection loss can be noisy or biased, leading to an inaccurate canonicalization and, in turn, degraded downstream learning. To quantify this effect, we vary $K$ across all environments while keeping the rest of the training pipeline fixed, and evaluate the resulting performance. Figure 9 shows a clear dependence on action-sample coverage. With a very small $K$, our method can underperform the standard reward-robust baselines, consistent with an unreliable estimate of the log-sum-exp term. This is particularly pronounced in high-dimensional continuous control tasks (e.g., Hopper), where a handful of action samples (e.g., $K = 4$) provides a poor approximation. As $K$ increases, the projection estimate becomes more accurate and performance improves accordingly, eventually recovering and typically exceeding the baseline. Overall, this ablation highlights a practical tradeoff: the gains from shaping-aware projection are realized when the projection loss is estimated with sufficient action coverage, at the cost of additional sampling in the projection-learning step.

**Ablation on canonicalization choices** To isolate the contribution of our $\Omega^*$-aligned canonicalization, Table 6 compares several projection choices under reward perturbations. First, we include a random $\Phi$ variant, which injects an arbitrary PBRS shaping term into the reward. This variant does not yield meaningful robustness improvements, indicating that the gains of our method do not come from simply adding an extra shaping-like component, but from learning a canonicalization that removes PBRS redundancy in a principled way. We also consider an $L^2$ projection that performs orthogonal projection regardless of the uncertainty-set geometry. This variant still removes PBRS-equivalent directions and thus improves robustness relative to the baseline, confirming that redundancy removal alone is beneficial. However, the $\Omega^*$-aligned projection performs better in most cases, suggesting that matching the canonicalization to the robustness geometry provides additional gains under reward perturbations beyond what a generic $L^2$ projection achieves.

