# OpenReview forum: "Behavior-Discriminative Reward Shaping for Reward-Robust Reinforcement Learning"
_ICML.cc/2026/Conference — Submitted to ICML 2026_

### Official Review · Reviewer_AWQZ · 2026-03-10

**Soundness:** 3
**Presentation:** 3
**Significance:** 2
**Originality:** 3
**Overall Recommendation:** 3
**Confidence:** 3

**Summary:**

This paper proposes a framework called Shaping-Aware Reward-Robust RL, which incorporates Potential-Based Reward Shaping (PBRS) equivalence into reward-robust reinforcement learning. The key observation is that standard uncertainty sets based on reward discrepancies contain many redundant rewards that are behaviorally equivalent, and this redundancy can hurt optimization. By projecting rewards to a canonical representative within PBRS equivalence classes, the method removes this redundancy while still preserves the optimal robust value. Overall, the idea proposed in this paper is well-motivated.

**Compliance With Llm Reviewing Policy:**

Affirmed.

**Ethical Review Concerns:**

None.

**Key Questions For Authors:**

1. How sensitive is the learned canonicalization to the quality or diversity of data used during the "lightweight projection-learning step"? If the data is limited or biased, will the projection still work well?
2. Can this approach interact with reward-learning settings (e.g., RLHF), where the “true reward” evolves over time?
3. Can you explain the impact of the approach on environments with different complexity levels, e.g., different state-action space sizes?

**Limitations:**

The projection step depends on approximating a shaping potential, which may introduce bias. Also the experiments focus on mid-scale continuous control tasks only. Broader validation on, for example, Atari or real robotics setting are missed but encouraged to have, which make it hard to judge how general the method is.

**Strengths And Weaknesses:**

- soundness
Strength: The theoretical part is a strong point. Authors provide formal proofs that the proposed projection preserve the optimal robust value, and also establish convergence guarantees for the practical algorithm.
Weakness: The reliance on robustness-regularization correspondence may limit the approach to specific types of uncertainty sets, for example entropy-based or KL-based ones. It is not clear how the method will perform if the uncertainty set has different structure.
- presentation
Strength: The paper is well-structured. The main idea of the work can be followed without too much difficulty.
Weakness: some of the complex mathematical derivations and implementation details are located in appendix, which makes the main text feel a little bit dense in some parts. I think authors can try to bring at least the key intuition of those derivations back to main paper, even if only briefly. Otherwise, it is rather difficult to interpret them.
- significance
Strength: This work addresses a meaningful gap in robust RL, especially the redundancy problem caused by PBRS invariance part.
Weakness: The empirical gain varies in magnitude across different MuJoCo environments. In some environments, the improvement is quite minor (Fig. 3). More convincing results are a must.
- originality
Strength: The idea of constructing uncertainty sets over PBRS equivalence classes is novel.
Weakness: The paper relies heavily on existing PBRS machinery and convex conjugacy results from prior work. So the originality is more in the integration and the new perspective rather than completely new technical primitives. The experiment results are not quite convincing.

---

> ### Author Rebuttal · Authors · 2026-03-30
>
> Dear Reviewer AWQZ:
>
> We sincerely thank you for the thoughtful and detailed feedback. Below, we address your concerns point by point.
>
> Q1: Limited to specific types of uncertainty set
>
> A1: We agree that the robustness-regularization correspondence limits Algorithm 1 to uncertainty sets that satisfy the convex-conjugate structure. Without this, Theorem 4.1 no longer applies, and Algorithm 1 cannot be used directly. We acknowledge this as a limitation of the current work. However, we emphasize that many widely studied uncertainty sets in reward-robust RL do satisfy this correspondence, so our method still covers a broad class of settings considered in the literature. Moreover, even when the uncertainty set does not admit this structure, the projected max-min problem can still be solved using standard methods. In particular, Figure 2 shows a grid-world experiment with an $\ell_p$-norm uncertainty set, where we directly optimize the projected robust objective and obtain stronger robust solutions. Developing efficient algorithms for more general uncertainty sets is an important direction for future work.
>
> Q2: More convincing results
>
> A2: We agree that our current main experiments focus on continuous-control benchmarks; however, they already span a meaningful range of task complexities. In particular, Hopper and Reacher have the smallest state-action dimensions (state: 11/10, action: 3/2), Walker2d and HalfCheetah represent a mid-scale regime (state: 17, action: 6), and Humanoid is substantially higher-dimensional (state: 348, action: 17). As shown in Table 1, our method improves across all of these settings. The gains are especially pronounced in the mid-scale environments, e.g., 107.7% on HalfCheetah for the KL setting. For the relatively smaller and larger environments, the improvements are moderate but still consistent. Notably, even on the highest-complexity Humanoid task, our method still achieves a 12.2% improvement in the KL setting, suggesting that the benefit of our method persists beyond low-dimensional problems. We also emphasize that our experimental setting follows the convention in the theoretical robust RL literature, which is typically limited to toy or relatively small environments.
>
>
>
> To evaluate generalization beyond continuous-control tasks, we additionally test our method on Atari. Specifically, we consider Seaquest and Pong, both of which have 18 discrete actions and high-dimensional visual observations of size 210x160x3. We use lr $2.5e^{-4}$ and batch size 64 for both tasks and train Seaquest with 10,000,000 timesteps and Pong with 1,000,000 timesteps. We also extend the evaluation to a more challenging simulated robotics benchmark, Adroit Pen from Gymnasium-Robotics (state: 44, action: 23). We train this environment with 2,000,000 timesteps using lr $3e^{-4}$ and batch size 64.
>
> The results are:
>
> Seaquest: 604.00$\pm$68.59 vs 832.00$\pm$48.99 (Entropy), 616.00$\pm$39.19 vs 840.00$\pm$29.39 (KL)
>
> Pong:  14.38$\pm$2.15  vs 16.13$\pm$1.36 (Entropy), 15.24$\pm$1.35 vs17.88$\pm$1.74 (KL)
>
> Adroit Pen: 4239.94$\pm$114.38 vs 5085.26$\pm$112.20 (Entropy), 4434.22$\pm$68.89 vs      6087.30$\pm$63.68 (KL)
>
> Across all cases, our shaping-aware method consistently outperforms the standard reward-robust baseline, supporting the broader applicability of the framework across different state/action dimensions and task domains.
>
> Q3: Projection bias
>
> A3: Please refer to our response to Reviewer ETmK, Q1.
>
> Q4: Sensitivity to data coverage
>
> A4: Our method is indeed sensitive to the quality of the projection-loss estimate, especially in the continuous-action MuJoCo setting where the projection objective is approximated by Monte Carlo action sampling. We explicitly study this in Appendix D.5 by varying the number of sampled actions $K$. As shown in Figure 9, when $K$ is small ($K=4$), our method can underperform the standard reward-robust baselines due to an unreliable estimate of the projection objective. And as $K$ increases, the projection estimate becomes more accurate and performance improves and exceeds the baseline. Therefore, the gains from our method are realized when the projection objective is estimated with sufficient data coverage.
>
> Q5: Interact with reward-learning settings
>
> A5: Thanks for raising this interesting question. Our paper is formulated in the classical reward-robust RL setting, where training is performed using a fixed proxy reward, while the unknown true reward is assumed to lie within an uncertainty set. Our method aims to learn a conservative policy that performs well under the worst-case. From this perspective, even in settings where the true reward may evolve over time, our approach remains meaningful as long as the current true reward is contained in the uncertainty set. If the proxy reward itself is updated over time, Algorithm 1 can in principle be re-applied at each reward-update stage.

---

> > ### Author Rebuttal · Reviewer_AWQZ · 2026-04-03
> >
> > Thank you for the detailed rebuttal. I still feel that the paper's contribution may be limited, considering the comments from reviewer 41gZ and the limitations that the authors agreed. Hence, I would like to keep the score unchanged.

---

> > > ### Author Response · Authors · 2026-04-04
> > >
> > > Dear reviewer AWQZ,
> > >
> > > We sincerely thank the reviewer for the response, and we are glad that we have been able to fully address your concerns. We would like to briefly restate the main contributions of the paper as follows:
> > >
> > > 1. Our paper identifies a redundancy issue in existing reward-robust RL formulations: when the uncertainty set is defined directly in the raw reward space, it can contain a large subspace of rewards that are behaviorally equivalent under PBRS. To the best of our knowledge, this is the first work to examine robust RL uncertainty sets from a PBRS perspective, and it provides a new viewpoint for constructing uncertainty sets that are more behaviorally meaningful.
> > >
> > > 2. We propose the shaping-aware reward-robust RL problem, in which the uncertainty set is projected onto canonical representatives within each PBRS equivalence class to remove this redundancy. We also develop a practical Algorithm 1 to solve the resulting shaping-aware problem by extending the robustness-regularization correspondence to the projected uncertainty set. We agree that our theoretical development builds heavily on PBRS invariance and convex-conjugate tools from prior work, and that Algorithm 1 is not an entirely new technical primitive. However, we believe this integration is well motivated by our PBRS-based insight into redundancy, and that convex-conjugate duality is the natural tool for turning this insight into a practical algorithm. While a completely new technical primitive would of course be valuable, we believe that a careful and principled use of existing tools to solve a new problem is itself a meaningful contribution.
> > >
> > > 3. Empirically, we show that our method consistently outperforms standard reward-robust RL across multiple task domains, including continuous-control tasks, high-dimensional visual game tasks, and robotic settings. We also evaluate across tasks of varying complexity, from low-dimensional state-action spaces to substantially higher-dimensional ones, which we believe provides convincing evidence for the value of the proposed projection method.
> > >
> > > Regarding limitations, we agree with Reviewer 41gZ that our main convergence theory relies on a strong convexity assumption that generally does not hold for neural-network parameterizations, and this is indeed a limitation of the current theory. At the same time, we would like to emphasize that this is common in RL research: theoretical analyses are often developed under stronger and more tractable assumptions than those satisfied by the neural-network implementations used in practice. Much of RL theory is still built around tabular, linear, or otherwise structured function classes, whereas practical success often depends on deep neural networks, for which comparable end-to-end guarantees are much harder to establish. We therefore view our current theory as a simplified analysis that highlights the core mechanism and clarifies the role of projection, and we do not believe that the lack of strong convexity for neural networks should be viewed as a central weakness that undermines the overall contribution of the paper.

---

### Official Review · Reviewer_41gZ · 2026-03-11

**Soundness:** 2
**Presentation:** 2
**Significance:** 3
**Originality:** 3
**Overall Recommendation:** 3
**Confidence:** 4

**Summary:**

This paper studies behavioral redundancy in the uncertainty set under the setting of reward-robust reinforcement learning. The authors complain that existing methods usually define uncertainty in the reward space, but reward differences don't always reflect behavioral differences. Based on the invariance of potential-based reward shaping, the aythors propose to construct a projected uncertainty set containing only behaviorally distinct rewards by performing a canonical projection over PBRS equivalence classes, and prove that under this projection, the optimal robust value of the original reward-robust objective is preserved. Furthermore, the authors leverage the connection between robustness and regularization to convert the projected robust objective into regularized RL, and provide an algorithm that approximates the projection by learning the potential function. Experiments in discrete and continuous control environments demonstrate good performance.

**Compliance With Llm Reviewing Policy:**

Affirmed.

**Final Justification:**

The core issue is that the paper is presented as a theory-driven contribution, but the most important theoretical claims are still not fully closed. In particular, Theorem 4.1 remains insufficiently justified in the projected setting, and the rebuttal does not fully establish the precise conditions under which the robustness-regularization correspondence continues to hold after projection. In addition, the gap between the exact projection studied in theory and the learned approximate projection used in the algorithm remains substantial. The authors’ clarification that the current method should be viewed as a “theoretically inspired approximate implementation” confirms this gap rather than resolving it. Finally, the convergence analysis still relies on a strong convexity assumption that is unrealistic for neural networks, limiting the practical value of the guarantee.

Overall, while the rebuttal improves clarity, it does not fully resolve the main concerns about theoretical rigor and theory-practice consistency, so I keep my original score.

**Key Questions For Authors:**

- Please strictly clarify and unify the definition of Ω∗. The current definition in the main text is inconsistent with its later usage as a scalar discrepancy.

- The step in Theorem 3.3 stating that "robust value is preserved after projection" needs more rigorous explanation. Lemma 3.1 only shows that returns differ by a strategy-independent constant, but the main text does not clearly show how this constant is strictly eliminated in the final max-min objective.

- What conditions does Theorem 4.1 rely on? Why does the robustness-regularization correspondence still strictly hold in the projected setting? This is the most critical bridging result in the paper, but the explanation in main text is clearly insufficient.

- What is the relationship between the exact projection in theory and the learned approximate projection in the algorithm? The theoretical proofs concern properties of the exact projection, but the algorithm implements an approximate projection. The authors need to explain whether this gap affects the main conclusions or at least clarify that the current algorithm should be interpreted as an approximate implementation inspired by theory.

- Existing experiments have not proven that performance improvements actually come from the Ω∗-aligned canonicalization proposed in this paper. Can the authors add simpler reward preprocessing baselines, such as reward normalization, moving-average baseline subtraction, or other non-PBRS-aware canonicalization, to more directly demonstrate the necessity of the proposed method?

**Limitations:**

It is recommended that the authors more explicitly discuss:

- The theory depends on PBRS equivalence relations and fixed initial distribution settings.
- There is approximation error between exact projection and learned projection.
- Current convergence analysis relies on strong convexity assumptions that are unrealistic for neural networks.
- When reward misspecification is not PBRS-type redundant, the method’s benefit may be limited.

**Strengths And Weaknesses:**

## Strengths

- The paper focuses on the construction of the uncertainty set in reward-robust RL, which is a fundamental problem in robust RL. The observation that "reward discrepancy does not equal behavioral discrepancy" addresses a real gap in existing modeling.

- Using PBRS equivalence classes to construct a behavior-discriminative uncertainty set, and further compressing the original reward space via canonical projection, is a novel theoretical perspective. It reflects originality in problem characterization and theoretical organization.

- Starting from PBRS invariance, the paper introduces projected uncertainty sets, canonical representatives, and robust value preservation, then further connects to regularized RL. The overall logical chain is coherent, and the main design is clear.

## Weaknesses

- The definition of Ω∗ is inconsistent with its later usage, which directly affects the mathematical validity of the uncertainty set, projection objective, and related theorems. For a theory-driven paper, this is a serious issue.

- Theorem 3.3 ("robust value is preserved after projection") and Theorem 4.1 ("projected robust objective is equivalent to regularized RL") are the most important results, but the main text does not fully elaborate the required conditions or intermediate steps, reducing theoretical persuasiveness.

- The theoretical analysis assumes exact projection, but the algorithm implements an approximate projection. The paper does not clearly explain whether this approximation error affects the main conclusions, so the current presentation appears more like an "implementation inspired by theory" rather than a rigorously closed methodological proof.

- The convergence analysis in the appendix relies on a strong convexity assumption of the projection loss with respect to the potential parameters, which the authors themselves admit usually does not hold for neural networks. Therefore, there is a clear theory-practice gap between the convergence guarantee and the actual deep learning implementation, weakening the credibility of the theoretical part.

---

> ### Author Rebuttal · Authors · 2026-03-30
>
> Dear Reviewer 41gZ:
>
> We sincerely thank you for the thoughtful and detailed feedback. Below, we address your concerns point by point.
>
> Q1: Inconsistency in the definition of $\Omega^\star$
>
> A1: We sincerely thank the reviewer for pointing out this inconsistency. In fact, in Section 2 (Notation paragraph), $\Omega^\star$ should be defined as $\Omega^\star(y)=\max_{a\in C}{\langle a,y \rangle-\Omega(a)}$, so that $\Omega^\star$ is used consistently throughout the paper as a scalar discrepancy. We will revise this in the final version.
>
> Q2: Eliminate the constant in Lemma 3.1
>
> A2: We agree that Lemma 3.1 gives equality up to a constant $C(\Phi, \mu_0)=-E_{s_0 \sim \mu_0}[\Phi(s_0)]$, which depends on the potential function $\Phi$ and the initial state distribution $\mu_0$. Theorem 3.3 is exact when $C=0$. Theoretically, this constant can always be eliminated by introducing a common start state $s_0$ before the original initial states and setting $\Phi(s_0)=0$, a standard normalization also used in prior work. In practice, we enforce this condition by adding an extra loss term $E_{s_0}[\Phi(s_0)^2]$ in the projection learning step, as described in Appendix C.2 (page 30, lines 1603-1604). We will revise the paper to make this clearer in the final version.
>
> Q3: Insufficient explanation of Theorem 4.1
>
> A3: Theorem 4.1 relies on the fact that $\Omega^\star$ is the convex conjugate of $\Omega$ and on the Slater condition (Assumption B.5, i.e., the uncertainty set contains at least one feasible solution), which ensures that the robustness-regularization correspondence holds for the original uncertainty set. The correspondence continues to hold in the projected setting because the duality argument depends only on the uncertainty set being defined through $\Omega^\star$, and is agnostic to the particular reward domain. Projection only restricts the adversary to one canonical reward, without affecting the convex-conjugate structure. We will revise the main text to include these details.
>
> Q4: Approximation error between exact projection and learned projection
>
> A4: Please refer to our response to Reviewer ETmK, Q1. We will also add the clarification that the current algorithm is an approximate implementation inspired by theory.
>
> Q5: Other reward preprocessing baselines
>
> A5: To isolate the contribution of our $\Omega^\star$-aligned canonicalization, in Table 2 (Section 5) and Table 6 (Appendix D.5), we report the ablation results of additional reward preprocessing baselines, including random canonicalization (non-PBRS-aware, implemented by randomizing the parameters of the potential function $\Phi$) and an $L^2$-projection canonicalization (which selects, within each equivalent rewards, the representative with the smallest $L^2$ value instead of the smallest $\Omega^\star$ value). We show that these variants show worse performance compared to the proposed method. For instance, in the Walker2d environment, our method achieves a return of $-188.14\pm9.26$, while the random one achieves $-226.17\pm36.72$ and the $L^2$-projection achieves $-199.04\pm9.37$. These results demonstrate the necessity of the proposed $\Omega^\star$-aligned canonicalization method.
>
> Q6: Unrealistic strong convexity assumption for neural networks
>
> A6: We agree that the current convergence guarantee relies on a strong convexity assumption for the projection objective, which does not directly apply to the neural-network setting and remains a limitation of the current work. Theoretically, this assumption is used to ensure a well-behaved projection and support convergence to the desired optimum over the projection parameters $\varphi$. In practice, to improve the conditioning of the projection objective and facilitate optimization, we add a ridge term $\frac{\lambda}{2}|\varphi|^2$ to the projection loss, as described in Appendix C.2 (page 30, lines 1603-1604). Although the projection is still learned only approximately in this deep setting, this regularization helps stabilize the projection-learning process in practice [1] and leads to promising empirical performance in our experiments.
>
> [1] Zhang, Guodong, et al. "Three Mechanisms of Weight Decay Regularization." ICML.
>
> Q7: Limitations of relying on only PBRS setting
>
> A7: We agree that our theory builds on PBRS equivalence relations, and thus the benefit of the method may be limited when reward misspecification is not PBRS-type redundant. We argue that many widely used uncertainty sets in reward-robust RL literature, including $\ell_p$-norm, entropy-based, and KL-based uncertainty sets, do contain PBRS-type redundancy as shown in the paper. We also acknowledge that there may be settings where the redundancy is not of PBRS type. Extending our framework to handle other forms of redundancy, or more general behavior-discriminative uncertainty sets beyond PBRS, is not covered in the current paper and is an important direction for future work.

---

> > ### Author Rebuttal · Reviewer_41gZ · 2026-04-02
> >
> > Thanks for the response. Regarding the inconsistency in the $\Omega^*$ notation and the treatment of constant terms in Lemma 3.1, the authors have provided sufficiently direct clarifications. However, as the authors acknowledge,with respect to Q6, the current convergence proof relies on a strong convexity assumption that is unrealistic for neural networks; this is rightly identified as a limitation of the paper. Regarding Q3, the high-level explanation provided by the authors aids in grasping the underlying intuition, yet it still fails to fully elucidate why Theorem 4.1 holds in a strictly mathematical sense, or how the relevant conditions are preserved following the projection step. Regarding Q4, the authors now explicitly concede that the proposed algorithm should be interpreted as a "theoretically inspired approximate implementation". This indicates a gap between theory and algorithm persist. More importantly, the supplementary analysis of approximation errors provided by the authors remains contingent upon Assumption B.12 and does not genuinely resolve this concern. I will keep the original rating.

---

> > > ### Author Response · Authors · 2026-04-04
> > >
> > > Dear reviewer 41gZ,
> > >
> > > We thank the reviewer for the follow-up questions.
> > >
> > > We next derive a strictly mathematical proof of why Theorem 4.1 holds for projection.  First, by Theorem 3.3, the projected set preserves the robust objective:
> > >
> > > $\max_{\pi}\min_{\tilde r_{\text{core}}\in U_{\text{core}}} J(\pi,\tilde r_{\text{core}})=\max_{\pi}\min_{\tilde r\in U} J(\pi,\tilde r).$
> > >
> > > Second, by Theorem B.6, the duality holds for the original uncertainty set
> > >
> > > $\min_{\tilde r\in U} J(\pi,\tilde r)=J(\pi,r)-\lambda(\Omega(\pi)-\varepsilon)$
> > >
> > > Finally, Lemma 3.1 implies $J(\pi,r_{\mathrm{core}})=J(\pi,r)$ under the normalization $C=0$. Therefore,
> > >
> > > $\max_{\pi}\min_{\tilde r_{\text{core}}\in U_{\text{core}}} J(\pi,\tilde r_{\text{core}})=\max_{\pi}(J(\pi,r_{\text{core}})-\lambda\Omega(\pi)+\lambda\varepsilon)$
> > >
> > > Regarding Q4 and Q6, we want to emphasize that it is common in RL research to present theory under stronger, more tractable assumptions than those satisfied by the neural-network implementation used in experiments. In particular, much of RL theory is still built around tabular, linear, or otherwise structured function classes, while practical success often relies on deep neural networks, for which comparable end-to-end guarantees are much harder to obtain. For example, [1] derives a two time-scale convergence result under Assumptions 6-7, which do not hold for the deep neural network used in their experiments, but are introduced to illustrate the behavior of the proposed algorithm (page 12).
> > >
> > > Accordingly, our main theory and supplementary analysis should be interpreted as clarifying the role of projection error and explaining the underlying mechanism, while the neural-network implementation serves as a practical approximation that demonstrates empirical value.
> > >
> > > In addition, even without Assumption B.12, we can still estimate the gap between the empirical objective and the true objective as
> > >
> > > $L(\varphi)-\hat L(\varphi)=\mathcal{O}((\frac{p\log N + \log(1/\delta)}{N})^{1/2})$.
> > >
> > > Moreover, the same stationarity-error decomposition still applies, with the projection error contributing only to the third term.
> > >
> > > We also emphasize that, even without strong convexity, one can still analyze convergence under weaker conditions such as Hölderian error bounds [2] or the Polyak-Łojasiewicz inequality [3], which are widely used in the nonconvex optimization literature and are more realistic for neural networks. However, pursuing such an extension is beyond the scope of the current paper. Overall, we view our current theory as a simplified analysis that highlights the main idea, and the lack of strong convexity for neural networks should not be seen as a central weakness.
> > >
> > > [1] Macua, Sergio Valcarcel, et al. "Fully distributed actor-critic architecture for multitask deep reinforcement learning." The Knowledge Engineering Review 36 (2021): e6.
> > >
> > > [2] Jiang, Rujun, and Xudong Li. "Hölderian error bounds and kurdyka-łojasiewicz inequality for the trust region subproblem." Mathematics of Operations Research 47.4 (2022): 3025-3050.
> > >
> > > [3] Liao, Feng-Yi, Lijun Ding, and Yang Zheng. "Error bounds, PL condition, and quadratic growth for weakly convex functions, and linear convergences of proximal point methods." 6th Annual Learning for Dynamics & Control Conference. PMLR, 2024.

---

### Official Review · Reviewer_ETmK · 2026-03-15

**Soundness:** 3
**Presentation:** 4
**Significance:** 4
**Originality:** 4
**Overall Recommendation:** 5
**Confidence:** 4

**Summary:**

This paper addresses a redundancy in Reward-Robust RL: standard uncertainty sets constrain deviations in raw reward space, which encapsulates massive subspaces of rewards that are equivalent under Potential-Based Reward Shaping (PBRS). Because PBRS equivalent rewards induce identical policy rankings, adversarial optimization is heavily wasted on behaviorally irrelevant directions. The authors propose Shaping-Aware Reward-Robust RL, which projects candidate rewards onto a canonical representative within their PBRS equivalence class. By leveraging the duality between robust RL and regularized RL, the algorithm alternatively learns the projection potential network and updates the policy, narrowing the adversary's search space.

**Compliance With Llm Reviewing Policy:**

Affirmed.

**Key Questions For Authors:**

1. Given that your neural network approximation of $\\Phi\_\\varphi$ incorporates ridge regularization and gauge penalties, to what extent does this destroy the exact PBRS invariance guaranteed by Theorem 3.3, an you formally bound this constraint violation?
2. Figure 9 shows massive degradation if the action sampling $K$ is small. Since estimating continuous action log-partition functions suffers from the curse of dimensionality, how does this method practically scale to action spaces with $\>20$ dimensions?
3. How sensitive is the framework to the learning rate ratio between the policy optimizer and the projection optimizer?

**Limitations:**

Yes. The authors include adequate discussion in Appendix D on the computational costs and the sensitivity to data coverage during the projection estimate, acknowledging the trade-offs of Monte Carlo action sampling.

**Strengths And Weaknesses:**

### Strengths
* Pointing out that standard $L\_p$ and log-sum-exp reward balls contain massive amounts of behaviorally redundant PBRS variations is an insightful observation. Proposition 3.4 proves how this projection reduces the affine dimension of the adversary's search space.
* Extension of robustness-regularization duality to the projected uncertainty set (Theorem 4.1) is robust. It shifts the computational burden of projecting every candidate reward onto optimizing a single base reward projection instead.

### Weaknessses
* Theorem 4.1 relies on an exact projection operator. However, in Algorithm 1, the projection $\\Phi\_\\varphi$ is approximated via a neural network optimized on a finite minibatch. The authors must add a ridge penalty $\\lambda||\\varphi||^2$ and a gauge condition to make it tractable. Consequently, the actual algorithm optimizes an approximation that breaks exact PBRS invariance, meaning the adversary may still leak into and exploit behaviorally relevant directions. The paper lacks a theoretical bound on how projection approximation errors impact worst-case return guarantees.
* Implementing the projection in continuous action spaces requires estimating a log-partition function via Monte Carlo importance sampling. As demonstrated in Figure 9, the algorithm is intensely sensitive to the number of sampled actions $K$. If $K$ is too small, the projection becomes biased, causing the method to perform worse than the baseline. This imposes sample/computational burden that doesn't scale well to high-dimensional continuous action spaces.
* The convergence analysis (Theorem B.17) relies on two-timescale stochastic approximation where projection network $\\Phi\_\\varphi$ converges faster than policy. If the projection network lags, the policy chases a moving, non-canonical target, which can induce training instability.

---

> ### Author Rebuttal · Authors · 2026-03-30
>
> Dear Reviewer ETmK:
>
> We sincerely thank you for the positive, insightful, and constructive review. We greatly appreciate your recognition of our theoretical contribution as both insightful and robust. Below, we address the remaining concerns.
>
> Q1: Approximation error of projection
>
> A1: We thank the reviewer for pointing out this important gap. In fact, the gauge condition is only a normalization used to fix the policy-independent constant $C=0$ in Lemma 3.1, while the ridge penalty is introduced to improve optimization stability. Neither of these changes the PBRS invariance itself. The main source of deviation from the exact projector in Theorem 4.1, as the reviewer pointed out, is that the projection is learned approximately via a neural network on a finite minibatch.
>
> To quantify this approximation error, let
>
> $L(\varphi):=\mathbb{E}\left[l(\varphi)\mid \theta\right]$
>
> denote the population projection objective under the rollout distribution induced by policy $\pi_\theta$, where
>
> $l(\varphi)=\Omega^\star\bigl(r-F_{\Phi_\varphi}\bigr)$
>
> be the projection loss. Let $l(\varphi;\tau)$ denote its empirical estimate computed from a rollout sample $\tau\sim \mathcal{D}$, and define the empirical objective
>
> $\hat L(\varphi):=\frac{1}{N}\sum_{n=1}^{N} l(\varphi;\tau_n),$
>
> where $N$ is the number of samples. Let
>
> $\varphi^\star\in \arg\min_{\varphi} L(\varphi), \hat\varphi\in \arg\min_{\varphi} \hat L(\varphi),$
>
> and define the corresponding projected rewards
>
> $r^\star = r - F_{\Phi_{\varphi^\star}}, \hat r = r - F_{\Phi_{\hat\varphi}}.$
>
> Under this setup, one can show that with probability at least $1-\delta$
>
> $\|\hat r-r^\star\| = \mathcal{O}((\frac{p\log N + \log(1/\delta)}{N})^{1/4}),$
>
> where $p$ is the dimension of the projection network parameter, thus highlighting that the complexity of our projection does not scale with the size of the state-action space. The proof follows a standard two-step argument. First, we establish a uniform bound showing that the empirical objective $\hat L(\varphi)$ is uniformly close to $L(\varphi)$. Then, under Assumption B.12, this objective-gap bound can be translated into a bound on the projection error, yielding the final result.
>
> Theorem B.17 establishes the convergence of Algorithm 1 through two-timescale tracking; however, it does not quantify how finite-sample errors affect the worst-case return guarantee. To do this, we apply Theorem 8 of [1] by viewing the optimization of our shaping-aware objective as an instance of optimizing a general utility. Under this perspective, Algorithm 1 admits a stationarity guarantee of $\mathcal{O}(1/I+1/M+(\frac{p\log N + \log(1/\delta)}{N})^{1/4})$, where $I$ is the number of outer iterations and $M$ is the number of samples used for policy gradient. Notably, the error consists of three terms: the first two correspond to the standard errors incurred by policy gradient methods, while the third arises from the approximation error in the learned projection.
>
> We will include the detailed derivation in the final version.
>
> [1] Barakat, Anas, et al. On the Global Optimality of Policy Gradient Methods in General Utility Reinforcement Learning.
>
> Q2: Scale to high dimensional action space
>
> A2: To understand if our method can practically scale to action spaces with >20 dimensions, we extend the evaluation to a more challenging simulated robotics benchmark, Adroit Pen from Gymnasium-Robotics with 23-dimensional actions. We train with 2,000,000 timesteps using lr $3e^{-4}$, batch size 64, and sample $2^8$ actions. The results are: 4239.94$\pm$114.38 vs 5085.26$\pm$112.20 (Entropy), 4434.22$\pm$68.89 vs 6087.30$\pm$63.68 (KL). Despite the high-dimensional action space, our method still achieves meaningful improvements, demonstrating that it can scale effectively.
>
> Q3: Sensitivity to learning rate ratio
>
> A3: We thank the reviewer for this insightful question. We argue that the learning-rate ratio is not critical for the convergence of Algorithm 1. From a theoretical perspective, the role of this condition is to ensure that the inner projection problem is tracked sufficiently well relative to the outer policy update. In Theorem B.17, this is formalized through a diverging learning-rate ratio, which is the standard way to ensure that the inner variable remains close to its desired solution while the outer policy evolves. In practice, the key requirement is therefore to choose the ratio so that the inner problem is solved sufficiently to remain well tracked. A larger ratio can further improve the inner solution, but should not affect convergence as long as the resulting error remains controlled. Empirically, we observe stable convergence behavior and consistent performance gains under the current ratio, which support this interpretation. We also acknowledge that the ratio may affect convergence speed, and our current analysis does not explicitly characterize this dependence. We view this as an interesting direction for future work.

---

### Official Review · Reviewer_uNg1 · 2026-03-23

**Soundness:** 4
**Presentation:** 4
**Significance:** 2
**Originality:** 3
**Overall Recommendation:** 5
**Confidence:** 5

**Summary:**

The paper shows that some reward uncertainty sets in robust MDPs can be interpreted as potential-based reward shaping, decomposing the adversary's search space into equivalence classes under policy-invariance. Instead of searching over raw worst-case rewards, the adversary searches over prototypes of these classes. This is implemented via policy gradient with a parameterized potential function

**Compliance With Llm Reviewing Policy:**

Affirmed.

**Key Questions For Authors:**

I think this is a very crucial bit in the paper: "picks the least inflated reward among all PBRS-equivalent
variants, yielding a consistent and geometry-aligned representative across classes"

I'm not sure I understand what you mean in this sentence with "inflated reward", "geometry aligned".

**Limitations:**

See above comment regarding ablations and optimization aspects, ie. the fact that the return metric doesn't convey much information in my opinion.

**Strengths And Weaknesses:**

I think the intro is too long. While reading it, it was hard to concretely grasp what you meant, for example when talking about representatives. I found myself just waiting to get to the math. The related work in Appendix A should be more prominent and actually be part of the main paper. I'd suggest cutting the intro down and using the recovered space for related work. It would really help position the paper.

That said, the paper is pretty neat. If I try to distill it: you consider certain uncertainty sets and interpret them as potential-based reward shaping, e.g. uncertainty sets that say "optimize for the worst-case reward within some prescribed maximum distance from a base reward." You then show in Proposition 3.2 that such uncertainty sets admit a potential function representation. Because of this, you can talk about invariance and preservation of optimal policies. In the robust problem, where you have a min adversary embedded inside a max, the min player is effectively searching over a space of potential-based equivalent rewards. You then recognize that this space decomposes into equivalence classes, and that instead of exhaustively searching over raw worst-case rewards, you can search within each class by defining prototypes for each partition.

You end up implementing this via a policy gradient framework. You make an executive decision to pick as prototype the "smallest" element within the equivalence class induced by the shaping function, then update the policy by policy gradient. In essence, you parameterize the potential function, solve the inner min in the space of shaping parameters, and carry out the policy update over the shaped reward in the usual way.

So the way I understand this paper: you do reward-robust RL, but define the adversary over the space of potential functions. The reason this is interesting is that you can argue the method hops between equivalence classes, potentially making the optimization problem easier to solve.

While you start strong with the regularized MDP perspective, the actual details as to how you end up using it are again buried in the appendix. I suggest also finding a way to bring some of it in the main text.

I think the practical impact will be moderate. People may find these uncertainty sets useful for their simplicity and interpretability, and for the potential to piggy-back on the reward shaping literature. It may help the optimization problem, but would really play a complementary role to more optimization-focused methods rather than parameterization-based ones like this. The practical benefits from the experiments are hard to assess in my opinion. Higher return, sure, but that's a pretty empty metric. I would have liked to see a more intentional and structured ablation, really something about the benefit of hopping over equivalence classes. From a theoretical point of view however, it's very neat, certainly worth publishing. Very clean.

---

> ### Author Rebuttal · Authors · 2026-03-30
>
> Dear Reviewer uNg1:
>
> We sincerely thank you for the positive, thoughtful, and detailed review. We greatly appreciate your recognition that the paper is theoretically neat, your careful understanding of our theoretical contributions, and your support for acceptance. Below, we address the remaining concerns.
>
> Q1: The return metric doesn't convey much information
>
> A1: We agree with the reviewer that return alone is an indirect metric for our central claim, namely that the proposed method avoids spending optimization effort on PBRS-equivalent reward directions. While an improved return does demonstrate practical benefit, it does not by itself fully isolate the mechanism.
>
> To directly capture the benefit of hopping over equivalence classes, in our MiniGrid experiments, we report the fraction of each adversary update that lies in PBRS-invariant directions. As shown in Figure 2c, a substantial portion of the updates made by the standard reward-robust RL baseline are behaviorally redundant. From the perspective of identifying genuinely adverse behaviors, these updates are effectively wasted, and may cause the optimization to stabilize prematurely at PBRS-equivalent rewards that provide little real progress in decreasing the worst-case value.
>
> For the more complex MuJoCo experiments, we cannot report the exact same measurement and eventually rely on the return to indirectly support our theory. To somehow show that the improvements do come from solving the adversary over the space of potential function using our projection method, we include an ablation study in Table 2 and Table 6 (Appendix D.5). In particular, we compare against variants that use random potential functions (implemented by randomizing the parameters of the potential function $\Phi$), which perform substantially worse than our full method. This shows that the gain does not come from adding an arbitrary shaping term, but from explicitly optimizing the projection to remove PBRS-equivalent directions in a principled way. We agree that additional measurements would further strengthen the presentation, and we view the current results as an important first step toward behavior discriminative uncertainty sets in robust RL that reflect behavioral distinctions rather than raw reward differences.
>
> Q2: Confusion regarding "inflated reward" and "geometry aligned".
>
> A2.  We thank the reviewer for pointing out this confusion. By “inflated reward,” we mean that within each PBRS-equivalence class, different representatives can have different magnitudes when measured by $\Omega^\star$, and our projection selects the one with the smallest $\Omega^\star$ value. By “geometry-aligned,” we mean that both the uncertainty set and the projection are defined using the same discrepancy functional $\Omega^*$, so the canonical representative is chosen consistently with the geometry induced by the uncertainty set. We agree that the original phrasing is unclear, and we will revise this sentence into “Intuitively, for each PBRS-equivalence class, our method selects the representative with the smallest $\Omega^\star$ value, yielding a consistent canonical reward that is aligned with the discrepancy geometry induced by the uncertainty set.”
>
> In addition, we will revise the paper by shortening the introduction and integrating the related work into the main text to better position our contributions. We will also improve the presentation of the theoretical results by clarifying key assumptions and adding proof details for important results such as Theorem 4.1, to make the arguments more precise and persuasive. We thank the reviewer again for carefully reading our paper and for providing these helpful comments.

---

### Decision · Program_Chairs · 2026-04-30

**Decision:**

Reject

**Comment:**

This paper studies robust reward RL where rewards are defined within an equivalent class. This way, the uncertainty set is smaller than over all potential rewards. The paper further proposes a regularized form and policy gradient methods and conducted experiments. The reviews are divided with two reviewers giving 5 while the other two give 3. The novelty lies mainly in the equivalent class while the critics focus on the approximation error as the approximate projection requires a neural network, a lack of precise conditions under which the robustness-regularization correspondence continues to hold after projection, and analysis relying on strong convexity. The overall idea builds upon the PBRS machineary. The paper should carefully consider the reviewers comments in the revised version for more precise presentation.